# Automated identification of sedimentary structures in core images using object detection algorithms

Ammar J. Abdlmutalib[1*], Korhan Ayranci[1*], Umair Bin Waheed[1], Hamad D. Alhajri[1], James A. MacEachern[2], Mohammed N. Al-Khabbaz[1]

**1** College of Petroleum Engineering & Geosciences, King Fahd University of Petroleum & Minerals, Dhahran, Saudi Arabia, **2** ARISE, Department of Earth Sciences, Simon Fraser University, Burnaby, British Columbia, Canada

\* ammar.mohammed@kfupm.edu.sa (AJA); korhan.ayranci@kfupm.edu.sa (KA)

## Abstract

Manual interpretation of sedimentary structures in core-based analyses is critical for understanding subsurface geology but remains time-intensive, expert-dependent, and susceptible to bias. This study investigates the use of convolutional neural networks (CNNs) to automate structure identification in core images, focusing on siliciclastic deposits from deltaic, shoreface, fluvial, and lacustrine environments. Two object detection models—YOLOv4 and Faster R-CNN—were trained on annotated data-sets comprising 15 sedimentary structure types. YOLOv4 achieved high precision (up to 95%) with faster training and shorter inference times (3.2 s/image) compared to Faster R-CNN (2.5 s/image) under consistent batch size and hardware conditions. Although Faster R-CNN reached a higher mean average precision (94.44%), it exhibited lower recall, particularly for frequently occurring structures. Both models faced challenges in distinguishing morphologically similar features, such as mud drapes and bioturbated media. Performance declined slightly in tests involving previously unseen datasets (Split III), indicating limitations in generalization across varied core imagery. Despite these challenges, the results demonstrate the promise of deep learning for streamlining core interpretation, reducing manual effort, and enhancing reproducibility. This study establishes a robust framework for advancing automated facies analysis in sedimentological research and geoscientific applications.

## Introduction

Geosciences heavily rely on visual examinations of rock features such as outcrops, thin sections, cores, and scanning electron microscope images. Among these, core-based analyses are critical for understanding facies successions, depositional environments, and subsurface reservoir characteristics [1–4]. Facies analysis is a cornerstone of core-based studies, it depends on recognizing lithology, mineralogy,

**Data availability statement:** All data used in this study are publicly available through open-access institutional databases. A detailed list of dataset links is provided in the GitHub repository (https://github.com/amarjuma2010/Object-detection-for-sedimentary-structures) and as Supporting Information.

**Funding:** The author(s) received no specific funding for this work.

**Competing interests:** The authors have declared that no competing interests exist.

sedimentary structures, deformational features, bioturbation intensities, and fossils [5,6]. Sedimentary structures, in particular, provide key insights into sediment transport dynamics, energy conditions, and depositional processes [7,8], directly influencing reservoir properties at several scales [9,10].

While classical manual methods for describing sedimentary structures in well cores are labor-intensive and prone to human bias, they offer certain advantages. One key benefit is that sedimentologists can physically rotate cores, allowing for the examination of laminations and structures from multiple perspectives. This flexibility provides crucial insights into the true morphology of sedimentary structures, which may not be fully captured in static images. Moreover, the three-dimensional nature of sedimentary structures is often lost when interpreted from two-dimensional photographs, leading to potential misclassification. For instance, parallel lamination and low- or high-angle cross-lamination can appear similar depending on the core orientation and viewing angle, underscoring the importance of core manipulation during manual analysis. Automated systems depend on high-quality training datasets. However, a lack of sufficient or unbiased data can hinder their ability to generalize effectively, reducing accuracy in identifying sedimentary structures across varied geological settings.

The transition to automated methods, while addressing some limitations of manual descriptions, introduces its own challenges. Automation enables rapid and precise geological analysis, reduces human bias, and improves the consistency of feature identification. Convolutional Neural Networks (CNNs), a form of artificial neural network designed for image and video data [11], have revolutionized image classification tasks across fields, including geology. CNNs have been applied to lithofacies identification [12–19], bioturbation intensity identification [20–22], core-based minerals content prediction [23], core-based fracture detection [24], igneous rock types identification [25], and automated rock quality designation analyses [26]. These models minimize manual effort, enhance analysis accuracy, and modernize workflows.

CNN methods such as Mask-RCNN have enabled semantic segmentation and classification of core features, including lithology and texture [18]. Object detection, another CNN approach, goes beyond segmentation by identifying multiple objects with bounding boxes and confidence levels, offering systematic and precise evaluation [27,28].

In geology, object detection has identified features like fractures, mineral grains, and facies attributes [29–36]. However, sedimentary structures, a critical aspect of facies analysis, remain underexplored in automated approaches. The only dedicated work by Zhang [37] employed transfer learning to identify three structure types, achieving 91.11% accuracy but with limited scope. Other studies combine sedimentary structures with lithology but do not prioritize sedimentary structures as an independent focus [15,18].

This study addresses the gap by applying object detection to identify 15 sedimentary structure classes, representing diverse depositional environments. This represents the first extensive application of object detection for sedimentary structures, enabling accurate identification across previously unseen core-box images.

By reducing manual workload, minimizing human bias, and enhancing efficiency, this research provides a flexible and groundbreaking model for sedimentary structure analysis, offering significant advancements for industrial and academic applications.

## Data and methods

### Dataset

To develop a robust, automated tool for identifying sedimentary structures, we utilized three datasets comprising a total of 506 box core images of siliciclastic sedimentary rocks (Fig 1a). These datasets represent a wide range of depositional environments, including continental, transitional, and marine settings.

The first dataset (Dataset 1) includes 400 box core images covering various depositional environments (estuary, shoreface, offshore, and delta), different lithologies (mudstone, sandstone, and conglomerate), and a number of formations from Alberta, Canada (Glauconite Formation, Viking Formation, Belly River Formation, and Dunvegan Formation). These images have an average resolution of 2129×2969 pixels, with highly variable vertical and horizontal resolutions (ranging from 72×72–2400×2400 dpi) due to differences in lighting conditions, camera setups, and imaging protocols. The second dataset (Dataset 2) contains 100 publicly accessible box core images (ExxonMobil-SEPM Core Data – EPR Price River A and EPR Price River C Lower) representing fluvial, shoreface, and delta deposits from the Price River Formation in Utah (see Data Access in the GitHub repository) [38]. The images in this dataset have average dimensions of 3715x4700 pixels, with vertical and horizontal resolutions between 500x500 dpi and 1200x1200 dpi. The third dataset (Dataset 3) comprises six publicly available box core images of ephemeral to perennial lake siliciclastic deposits from the Permian

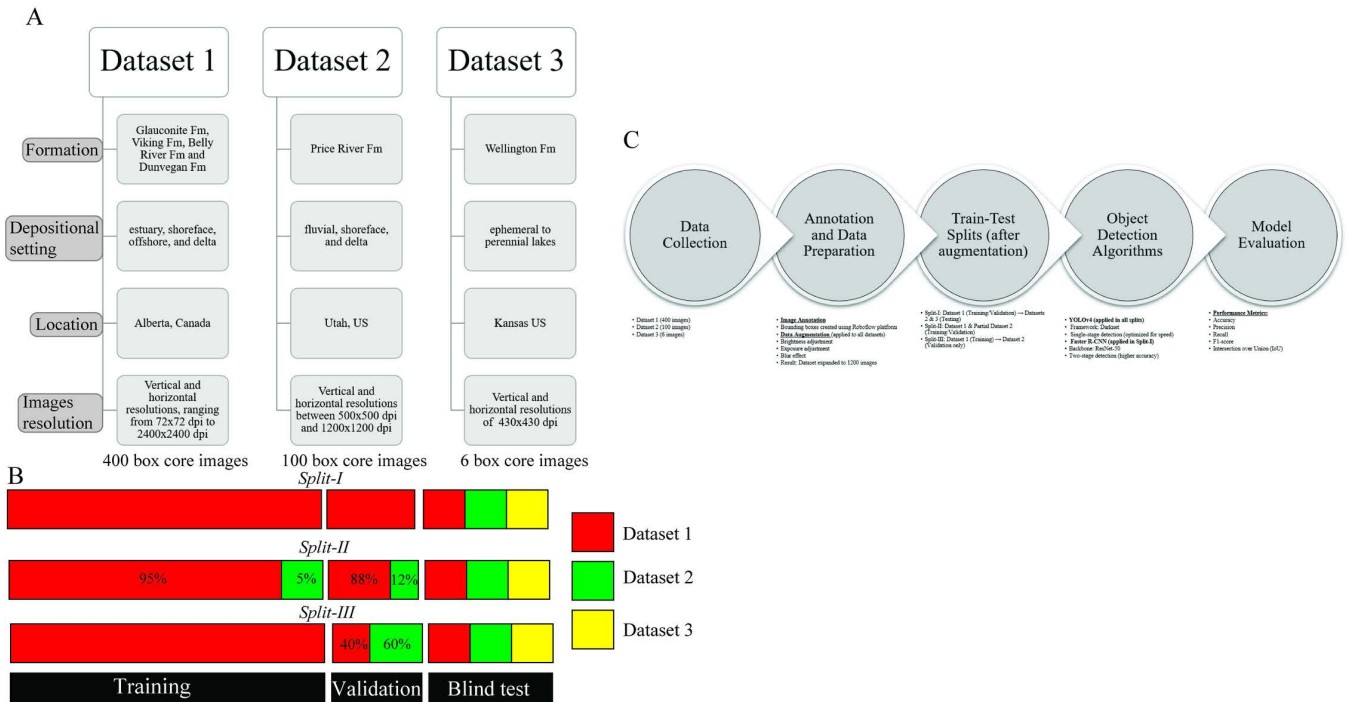

**Fig 1. Overview of datasets, train-test splits, and workflow used in this study.** (A) Summary of the three datasets, including geological setting, location, and number of core images. (B) Schematic of train-test-validation split configurations across the datasets. (C) Workflow diagram illustrating the methodological pipeline from data collection to model evaluation.

Wellington Formation [39]. These images have average dimensions of 1500x1941 pixels and a resolution of 430x430 dpi (see Data Access in the GitHub repository).

All bounding-box annotations were performed using Roboflow and were guided by a detailed labeling protocol rooted in sedimentological definitions. Critically, all annotations were reviewed by a team of domain experts—specifically, sedimentologists with over 15 years of experience in core description and facies interpretation. Expert validation focused particularly on distinguishing morphologically similar classes (e.g., mud drapes vs. bioturbated media), and discrepancies were resolved through consensus-based reannotation. This expert oversight was essential for ensuring high-quality, geologically meaningful labels in the training and test datasets.

## Methods

We conducted three train-test splits (Fig 1b). In each split, 70% of the entire box core images were assigned as training image data, 15% were used as validation and 15% were used as test dataset. In Split-I, Dataset 1 was partitioned into non-overlapping subsets, with 70% of the images used for training and 15% for validation. The remaining data from Datasets 2 and 3 was reserved for external testing. In Split-II, a subset of core images from Dataset 2 was incorporated into the training and validation data, with the majority of the training data (~95%) coming from Dataset 1 and a smaller portion (~5%) from Dataset 2. In Split-III, Dataset 2 was used solely for validation purposes. 60% of the validation images were taken from dataset 2. All datasets were ultimately used for blind testing. The images were categorized into 18 classes, which included 15 sedimentary structures and three background classes (section 2.3). Notably, the 13 sedimentary structures in Datasets 2 and 3 correspond to the most commonly represented structures in Dataset 1, reflecting their shared siliciclastic sedimentary settings. A comprehensive breakdown of class distributions across the training and test sets for each data split is provided in Supplementary S1–S3 Tables. These tables illustrate the presence and representation of each class after applying augmentation.

The methodological workflow, illustrated in Fig 1c, outlines the full data processing pipeline, from initial image acquisition and bounding-box annotation to dataset augmentation, train-test-validation splitting, and model training. Notably, this process incorporates extensive image augmentation (e.g., brightness, cropping, exposure, and blur adjustments) prior to dataset partitioning, ensuring improved model generalization. The diagram also emphasizes the use of structured training-validation-test configurations and the integration of both YOLOv4 and Faster R-CNN architectures for comparative analysis. Images from Dataset 1 and 2 were labeled with bounding boxes using the Roboflow online platform, which provides a range of preprocessing and augmentation options. Data augmentation was applied to expand the training dataset, thereby enhancing the model's performance and robustness. Specifically, our augmentation techniques increased the original dataset threefold (resulting in 1200 core box images) by adjusting brightness, exposure, and blur (Fig 2). These augmentations improved the model's flexibility, ensuring it could handle variations in camera settings, such as lighting and focus. Image augmentation were carried out prior to train-test splitting.

For object detection, we employed two algorithms: YOLOv4 (You Only Look Once, version 4) with the Darknet framework (Fig 3a) and Faster R-CNN with ResNet-50 as the backbone (Fig 3b). The applied hyperparameters are detailed in Table 1.

The YOLOv4-Darknet algorithm is renowned for its real-time object detection capabilities, offering a balance between accuracy and processing speed [28]. This algorithm utilizes convolutional neural networks (CNNs) for both feature extraction and object detection, making it particularly effective for identifying small- to medium-sized objects, aligning with the needs of our study [10]. We customized and trained YOLOv4 on our dataset, focusing on maximizing accuracy.

YOLOv4 streamlines the detection process by using a single CNN, as opposed to traditional methods that rely on multiple stages for object detection [3,27]. The process begins by segmenting the image into a grid, with each cell detecting objects within its bounds. The Darknet backbone CNN extracts features at various scales, which is essential for detecting objects of different sizes (Fig 3a). The model predicts multiple bounding boxes per grid cell, each with a confidence score

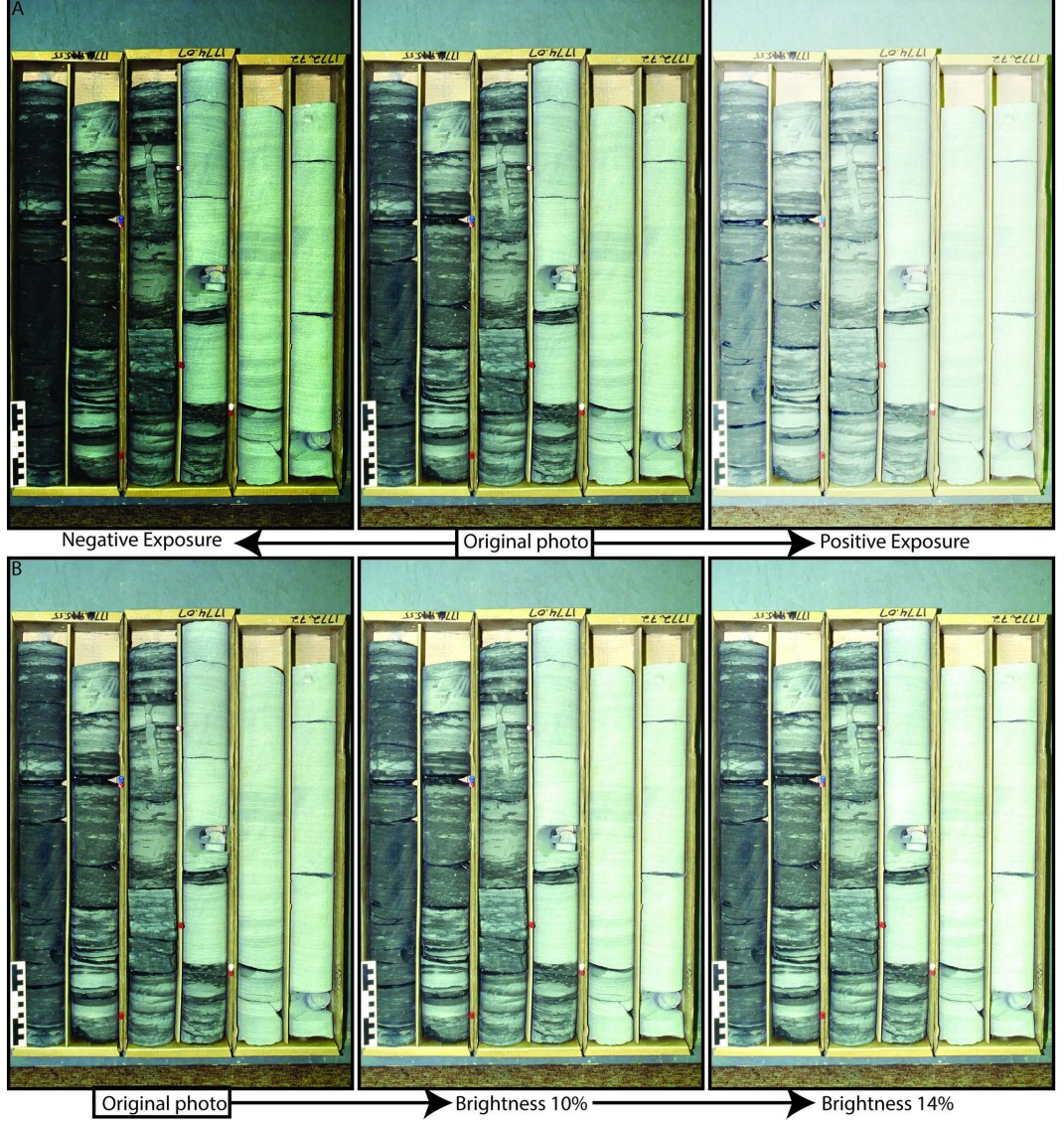

**Fig 2. Examples of data augmentation techniques applied to core images.** (A) Exposure adjustments showing under- and overexposed versions. (B) Brightness modifications applied incrementally to simulate lighting variation.

indicating the likelihood of an object's presence and the precision of the bounding box. It also predicts class probabilities for each box, effectively classifying the detected objects. YOLOv4 uses non-maximum suppression (NMS) to filter out overlapping boxes, retaining only those with the highest confidence scores.

Faster R-CNN with ResNet-50, employed in this study, is a deep learning model that integrates region proposal generation and object classification within a single framework (Fig 3b). ResNet-50, a 50-layers CNN, extracts hierarchical features from input images. The Region Proposal Network (RPN) then generates candidate object proposals by sliding over these feature maps, predicting objectness scores and refining bounding boxes. Using ROI pooling, the proposals are resized to a uniform size for consistent processing [40,41]. These refined proposals are then classified into object categories, with bounding boxes further adjusted for final detections, offering a balance of speed and accuracy.

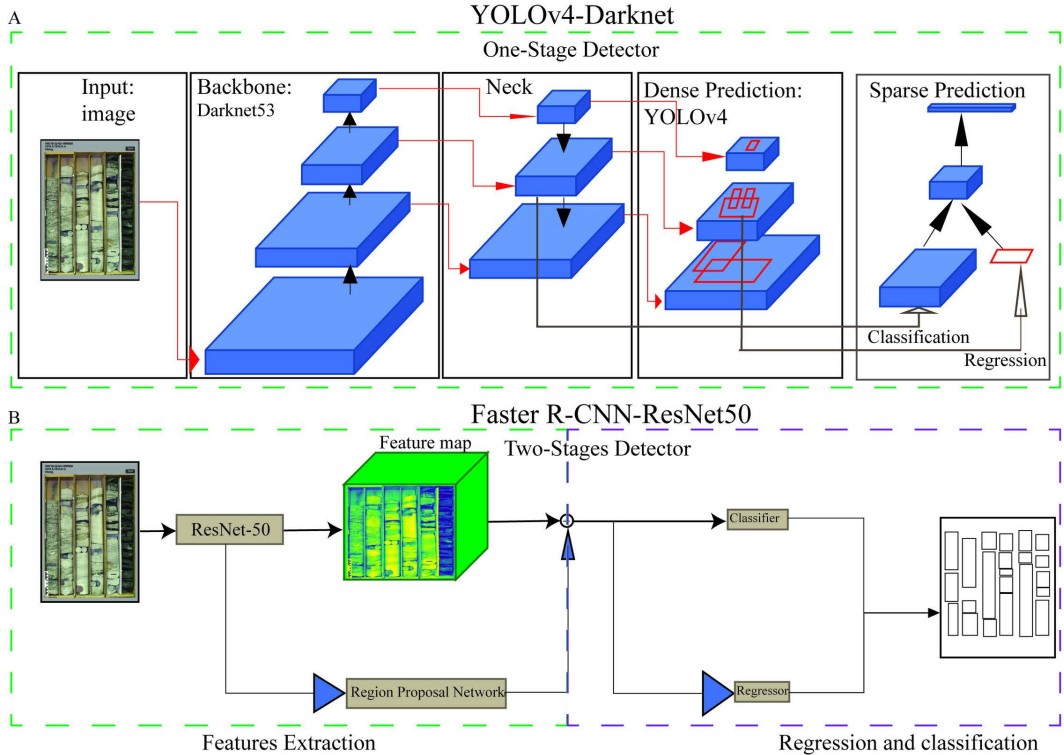

**Fig 3. Architecture of the object detection models used.** YOLOv4-Darknet and Faster R-CNN-ResNet50 are illustrated, with structural components modified after [27] and [40], respectively.

**Table 1. Key hyperparameters used in training YOLOv4-Darknet and Faster R-CNN-ResNet50 models, including batch size, learning rate, and optimizer settings applied in sedimentary structure detection tasks.**

|  | Hyperparameter | Value |
|---|---|---|
| **YOLOv4-Darknet** | Batch | 32 |
|  | Subdivision | 8 |
|  | Max_batch | 39000 |
|  | Steps | 31200, 35100 |
|  | Network size | 416*416 |
|  | Learning rate | 0.001 |
|  | Filters | 72 |
|  | Class weighting | Applied based on class distribution |
| **Faster R-CNN-ResNet50** | Batch size | 32 |
|  | Learning rate | 0.001 |
|  | Optimizer | Adam (torch.optim.Adam) |
|  | Number of epochs | 100 |
|  | Image input size | Padded |
|  | Collate function | Default |
|  | Class weighting | Applied during loss computation to address class imbalance |

While YOLOv4-Darknet is optimized for training speed and real-time detection with its single-stage architecture, Faster R-CNN with ResNet-50 offers higher accuracy through its more complex two-stage process. The choice between the two models depends on the specific application needs, such as speed versus accuracy. In our experiments, we evaluated the YOLOv4 model across all three splits, while Faster R-CNN was used only in Split-I to compare one-stage and two-stage detection algorithms.

To evaluate detection performance, we used five standard metrics: precision, which quantifies the proportion of correct predictions among all positive predictions; recall, which measures the proportion of correctly identified structures among all actual positives; F1-score, the harmonic mean of precision and recall; intersection over union (IoU), which quantifies the overlap between predicted and ground truth bounding boxes; and mean average precision (mAP), a comprehensive measure that averages the precision across all classes and detection thresholds [42]. For completeness, full mathematical formulas and further computational details are provided in the supplementary material (S4 Table).

The inference codes and model resources for this study are publicly available (see Access in the GitHub repository). The repository includes pre-trained model weights, inference scripts, and instructions for running predictions on new images and videos. This ensures full reproducibility of the object detection process applied in this study.

The inference code is capable of generating predictions on both images and videos, producing annotated results with detected bounding boxes, class labels, and confidence scores. The predicted videos demonstrating the model's performance are included in the attached GitHub repository.

## Sedimentary structures

We investigated a diverse range of depositional, erosional, and deformation sedimentary structures using core-box images. To ensure accurate identification and annotation, specific criteria were established for each structure based on their standard definitions. Among the most predominant features were individual mud drapes (Fig 4a), characterized by interbedded mud and thin sand beds within heterolithic intervals [43]. These structures are typical of tidal flat environments, often associated with flaser and wavy bedding, though rarely with ripple structures. Each mud drape in the dataset was individually labeled using bounding boxes for precise modeling.

Bioturbated muddy and sandy media were another focus of the study (Fig 4b and 4c). These classes were labeled based on the presence of burrowed media containing various ichnofossils, with examples ranging from sparsely to intensely bioturbated beds [44]. To maintain consistency, muddy and sandy media were differentiated by their overall sand-to-mud ratio. Massive sandstone and mudstone structures, which lack clear sedimentary lamination and\or bioturbation due to uniform texture or high sedimentation rates, were labeled as sandstone, mudstone, or conglomerate to reflect their lithological differences (Fig 4d-4f).

Other significant structures included parallel lamination (Fig 4g), which describes thinly laminated to very thinly bedded strata formed on stable flat beds, and current ripples (Fig 4h), found in silt to medium sand beds [7,45]. The latter, with wavelengths of 10–60 cm, excludes megaripples, which were absent in the core-box images. Low-angle lamination and cross-lamination were also annotated (Fig I and j), with a 10° angle threshold used to distinguish between them [44]. Additional features like fissile shale, rip-up clasts, soft-sediment deformation, and wavy bedding were labeled to capture their unique characteristics (Fig 4k-4o), though the latter two accounted for less than 0.15% of the training dataset. Rip-up clasts, gravel-sized fragments of clay or mud, result from erosive currents and are deposited in high-energy environments, such as channels or storm-influenced areas [46] (Fig 4l). Soft-sediment deformation structures form shortly after deposition in high-energy environments like turbidity currents or storm-driven zones (Fig 4n), reflecting rapid sedimentation and gravitational instabilities [47]. Wavy bedding (Fig 4o), characterized by alternating mud and rippled sand layers, forms from ripple migration in low-energy tidal settings [45]. This comprehensive labeling approach ensures robust data preparation for modeling sedimentary structures.

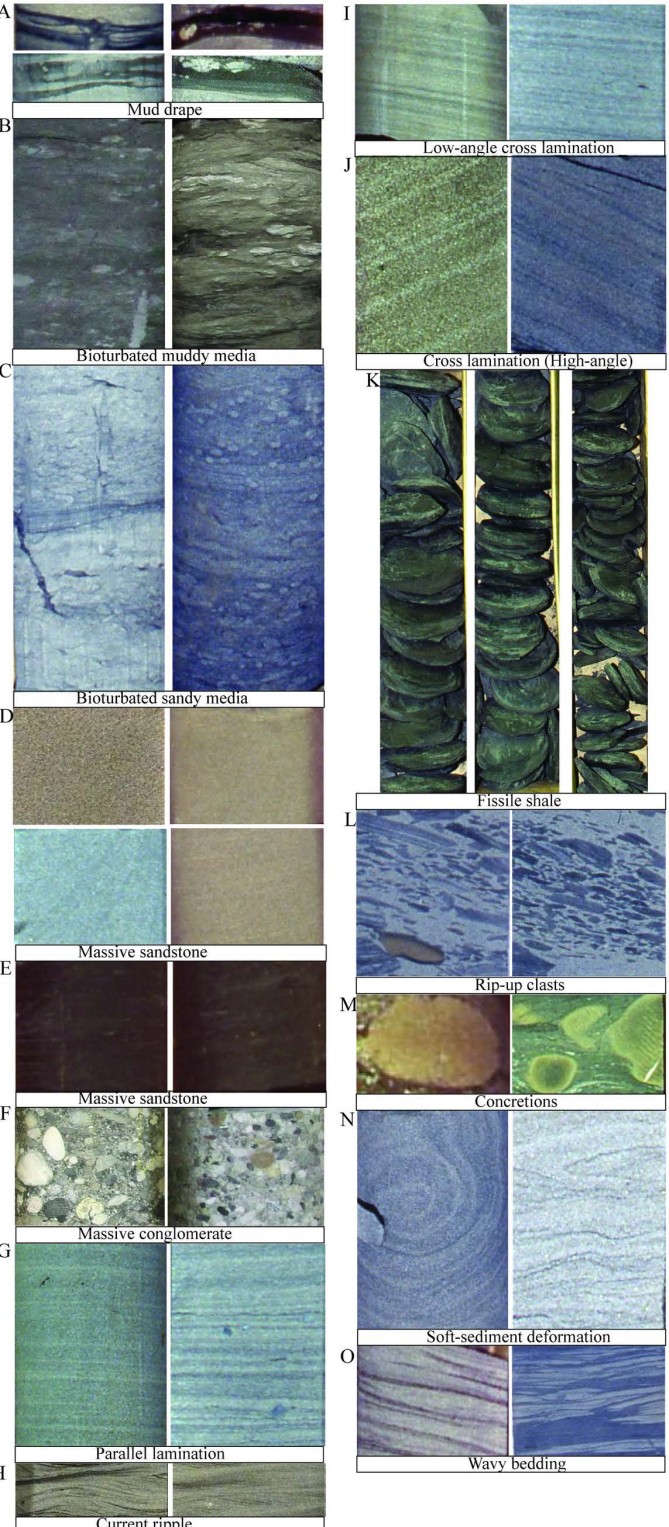

**Fig 4. Examples of annotated sedimentary structures in core box images.** Panels (A–O) illustrate key depositional, erosional, and deformational features used in training, including mud drapes, bioturbated media, laminations, and massive units.

## Results

### YOLOv4-darknet model performance evaluation

**Classification metrics.** Evaluation of the YOLOv4 model across three distinct training-validation-test splits revealed consistent high performance in Splits I and II, with a marked decline in Split III (Fig 5a–5b; Table 2). Precision remained strong at 0.95 and 0.93 in Split-I and Split-III, respectively, underscoring the model's ability to accurately detect sedimentary structures with minimal false detections. In contrast, Split III showed a noticeable drop in precision (0.83) and a slight decrease in recall (0.85), indicating increased false positives and a reduced true positive rate. This decline was also reflected in the F1-score, which dropped to 0.84, highlighting difficulties in balancing precision and recall for this data configuration. Intersection over Union (IoU), a metric for bounding box accuracy, was high in Splits I and II (80.84% and 84.41%) but dropped to 74.73% in Split III (Fig 5b), suggesting less precise localization. The mean average precision (mAP) followed a similar pattern, declining to 74.17% in Split III from above 92% in the other splits. Split-I emerged as the best-performing configuration, with the highest proportions of true positives and lowest false positives and negatives proportions relative to the entire predictions, followed closely by Split-I. Split-III yielded the lowest true positives and highest error rates (Fig 5c-5e). Loss curves confirmed successful convergence for all splits, though Split-I showed the smoothest training, while Split-III displayed more noise and slower convergence (Fig 5f-5h), likely due to increased data variability or complexity.

Class-wise analysis (Table 3 and S1-S3) revealed that more frequent classes—those with over 10% of bounding boxes, such as mud drapes, massive sandstone, and bioturbated muddy media—generally maintained high precision across all splits. Notably, despite its abundance (24.80%), mud drapes had the lowest precision among the dominant classes, likely due to their visual similarity to other features. In Split-II, their precision peaked at 89.29%. Less frequent classes, especially in Split-III, were challenging to classify accurately. For example, concretions and rip-up clasts, which have limited representation in the dataset, showed significant drops in precision in Split-III (37.61% and 61.6%, respectively). This semi-linear correlation between class frequency and detection accuracy in Split-III reflects real-world challenges where rare features are more difficult to detect reliably.

### Evaluation of predictions on test images

To evaluate the model's classification performance under complex geological conditions, we selected the most visually complex core box images from Dataset 1 as test cases (Figs 6–8). These images contain a broad spectrum of sedimentary structures, offering a rigorous test for model generalization. Overall, predictions across all three splits demonstrated good agreement with ground truth labels, although some errors were evident, particularly in the more variable scenarios. Among the splits, Split-I consistently showed the closest alignment with annotated labels, echoing the higher validation metrics reported for this configuration.

In the first test image (Fig 6), which includes mud drapes, bioturbated sandy media, and massive sandstone, all three models successfully detected most key features. Split-I predictions displayed strong alignment with the true labels (Fig 6b–6e). Notably, thin massive conglomerate beds were correctly identified (Fig 6f–6g). However, errors such as the misclassification of mud drapes as bioturbated sandy or muddy media were evident, particularly in Split-II and Split-III predictions. Split-II also failed to detect thin massive sand layers, underscoring the challenges posed by subtle textural transitions (Fig 6g).

The second image (Fig 7), featuring a more fragmented core and a broader diversity of structures, posed additional challenges. Nonetheless, model predictions largely corresponded with the true annotations (Fig 7b–7e). Split-I misclassified mud drapes as bioturbated sandy media, while both Split-II and Split-III confused current ripple cross-lamination with parallel lamination. Further, faint parallel lamination was occasionally misidentified as massive sandstone, and Split-I confused massive sandstone with low-angle cross-stratification or lamination (Fig 7f–7h).

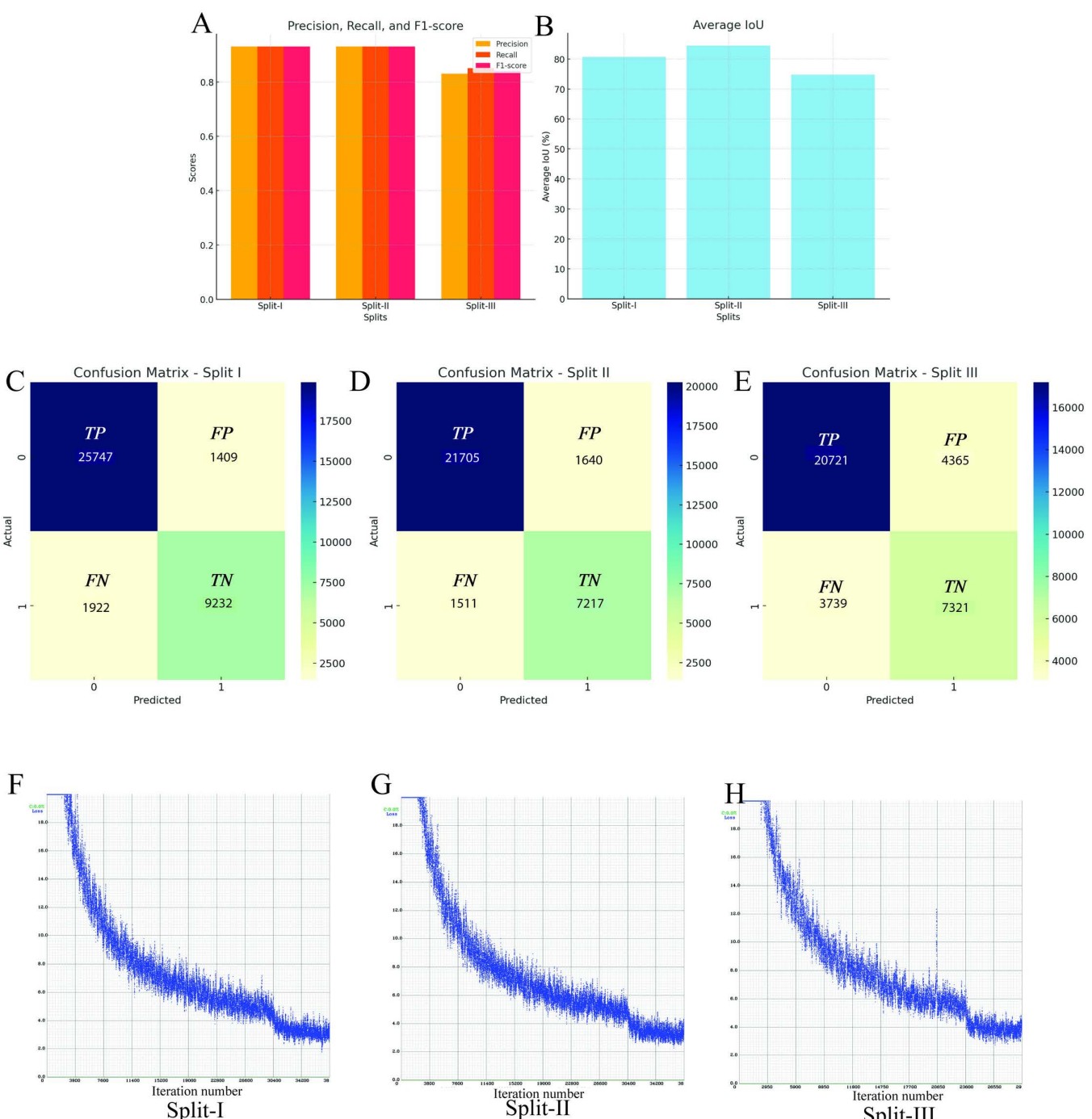

**Fig 5. Comparison of YOLOv4 performance across three data splits.** (A–B) Evaluation metrics including precision, recall, F1-score, and IoU. (C-E) Confusion matrices. (F-H) Training loss curves illustrating convergence behavior across splits.

**Table 2. Summary of YOLOv4 model performance across three data splits, including TP, FP, FN, precision, recall, F1-score, average IoU, and mean average precision (mAP).**

| Metrics | Split-I | Split-II | Split-III |
|---|---|---|---|
| Precision | 0.95 | 0.93 | 0.83 |
| Recall | 0.93 | 0.93 | 0.85 |
| F1-score | 0.94 | 0.93 | 0.84 |
| Average IoU | 80.84% | 84.41% | 74.73% |
| Mean average precision (mAP) | 92.82% | 93.11% | 74.17% |

**Table 3. Precision scores for each sedimentary structure class, showing class distribution and precision across Split-I, Split-II, and Split-III.**

| | | Average precision (%) | | |
|---|---|---|---|---|
| | Classes | Split-I | Split-II | Split-III |
| Sedimentary structure | Mud drapes | 82.39 | 89.29 | 84.71 |
| | Massive sandstone | 98.01 | 97.46 | 90.99 |
| | Bioturbated muddy media | 96.51 | 97.14 | 93.25 |
| | Massive mudstone | 96.95 | 96.42 | 92.92 |
| | Bioturbated sandy media | 98.04 | 98.19 | 94.01 |
| | Parallel lamination | 97.44 | 92.12 | 82.8 |
| | Low-angle lamination | 98.42 | 94.03 | 78.41 |
| | Massive conglomerate | 99.57 | 98.85 | 96.5 |
| | Cross stratification | 98.75 | 97.84 | 84.51 |
| | Current ripples | 84.06 | 91.77 | 54.71 |
| | Fissile shale | 99.99 | 93.53 | 90.86 |
| | Rip-up clasts | 98.93 | 98.11 | 61.63 |
| | Scattered pebble | 25.6 | 80 | 7.14 |
| | Concretions | 91.79 | 54.65 | 37.61 |
| | Soft-sediment deformation | 100 | 93.75 | 45.05 |
| | Wavy bedding | 100 | 100 | 100 |
| Non-rock | Broken pieces | 98.71 | 96.88 | 85.41 |
| | Empty | 99.95 | 99.83 | 95.97 |
| | Non-core | 98.45 | 99.26 | 32.79 |

In the third test image (Fig 8), which presented complex sequences including conglomerates and fine-grained inter-beds, predictions were again mostly consistent with ground truth (Fig 8b–8e). However, all models failed to detect a thin massive conglomerate, instead labeling it as massive sandstone—most notably in Split-III (Fig 8f–8g). Additional misclassifications included the confusion of massive sandstone with parallel lamination or low-angle cross stratification, and mud drapes being mislabeled as bioturbated muddy media, particularly in Splits II and III (Fig 8h).

These misclassifications primarily stem from overlapping morphological features or transitional textures between sediment types. Importantly, they are comparable to errors that might arise during manual interpretation, underscoring both the limitations and relative strengths of deep learning models in replicating human-level geological assessment.

## Blind test evaluation

To further assess model performance, a blind test was conducted using cropped core images that featured the most abundant and geologically significant sedimentary structures. All three split models demonstrated strong predictive

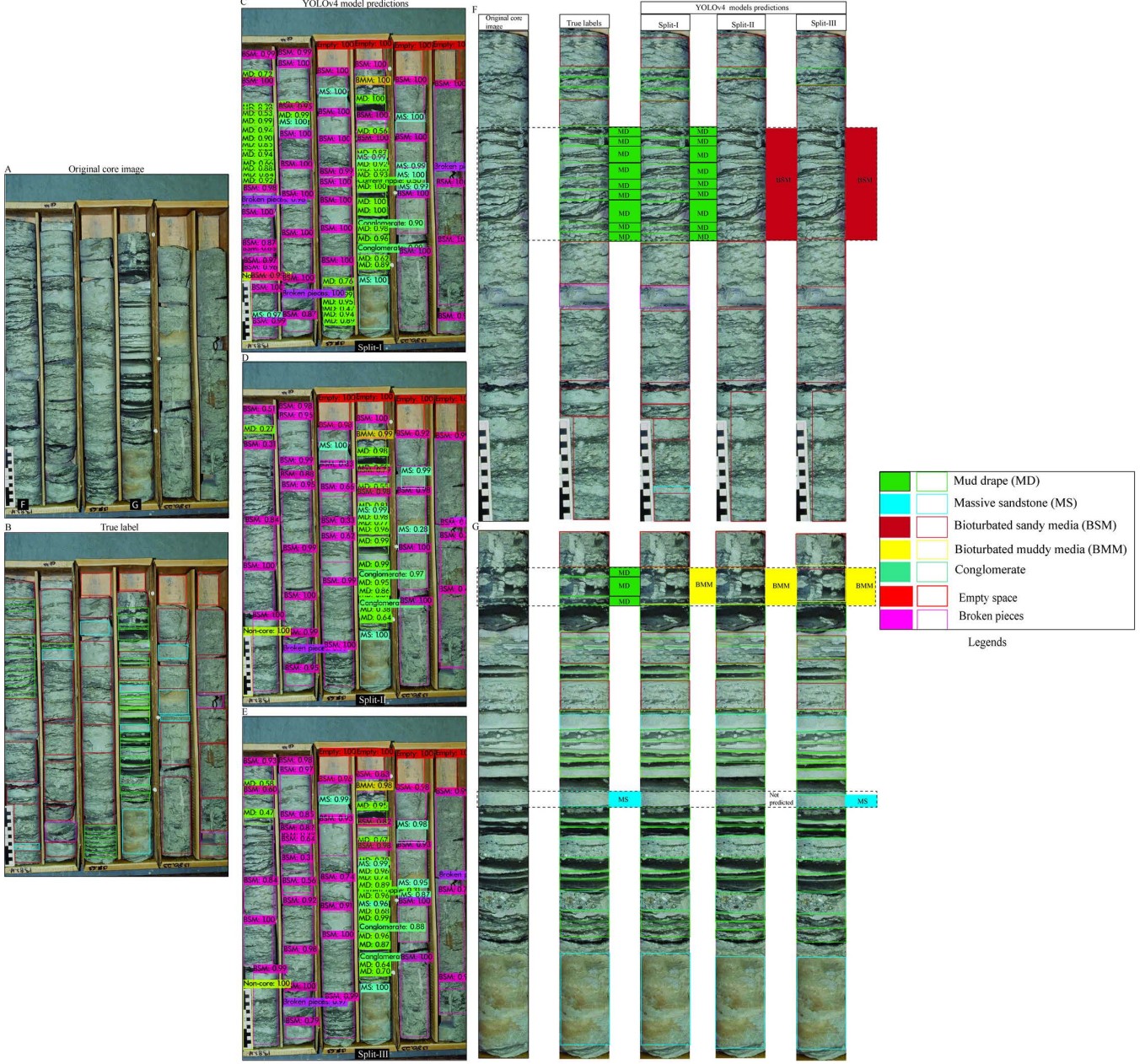

**Fig 6. YOLOv4 predictions on core images from Dataset 1 across all three data splits.** (A) Original image. (B) Ground truth annotations. (C–E) Model predictions per split. (F–G) Column-level comparison showing correct detections and misclassifications.

capabilities, accurately identifying bioturbated muddy media (Fig 9a), massive sandstone and mud drapes (Fig 9b), conglomerate (Fig 9c), parallel lamination (Fig 9d), bioturbated sandy media (Fig 9e), high-angle cross lamination (Fig 9f), and low-angle cross lamination (Fig 9g). However, some confusion was evident—particularly between mud drapes and bioturbated muddy media (Fig 9a–9b)—highlighting challenges in distinguishing structures with overlapping visual and textural characteristics.

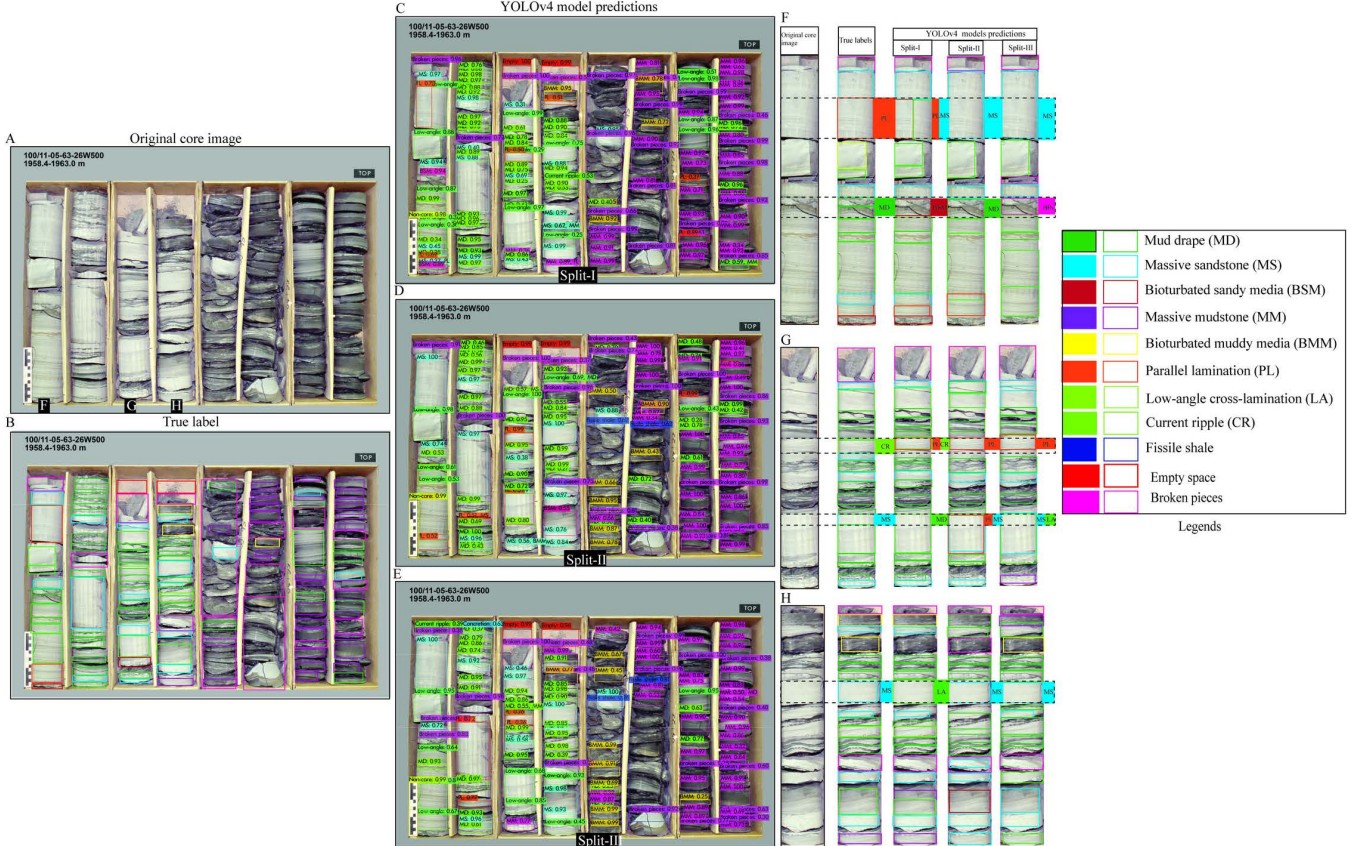

**Fig 7. YOLOv4 predictions for a core image with complex sedimentary structures.** (A–B) Original image and true labels. (C–E) Split-wise predictions. (F–H) Detailed column-wise comparisons highlighting variations in detection accuracy.

To compare model generalizability, a core box image from Dataset 2 was analyzed using the originally trained YOLOv4 model (Fig 10). This test image featured a variety of sedimentary structures, including mud drapes, massive sandstone, bioturbated muddy media, low-angle cross stratification, parallel lamination, and bioturbated sandy media, along with non-geological filler materials (Fig 10a). Among the splits, the Split-II model delivered the most accurate predictions, aligning closely with the ground truth labels (Fig 10b–10e). In contrast, Split-I misidentified bioturbated muddy media as mud drapes, while Split-III confused them with parallel lamination. Both Split-I and Split-III showed considerable misclassification between massive sandstone and bioturbated sandy media (Fig 10f–10g) and struggled to accurately delineate thick, parallel lamination beds (Fig 10h).

A second comparative test was performed using an image from Dataset 3, which included bioturbated sandy and muddy media, massive mudstone, mud drapes, and massive sandstone (Fig 11). Split-I again produced predictions that most closely matched the annotated labels (Fig 11b–11e). In contrast, Split-II and Split-III frequently misclassified bioturbated media as massive sandstone or conglomerate (Fig 11f). Split-II also showed broader misclassifications, labeling mud drapes as bioturbated sandy media, rip-up clasts, soft-sediment deformation, and wavy bedding. Similarly, Split-III misinterpreted mud drapes as bioturbated sandy media (Fig 11g).

These blind test results emphasize both the potential and the current limitations of the models. While overall performance was strong, consistent misclassifications among morphologically similar classes underline the need for enhanced feature discrimination, possibly through contextual stratigraphic data or texture-aware model enhancements.

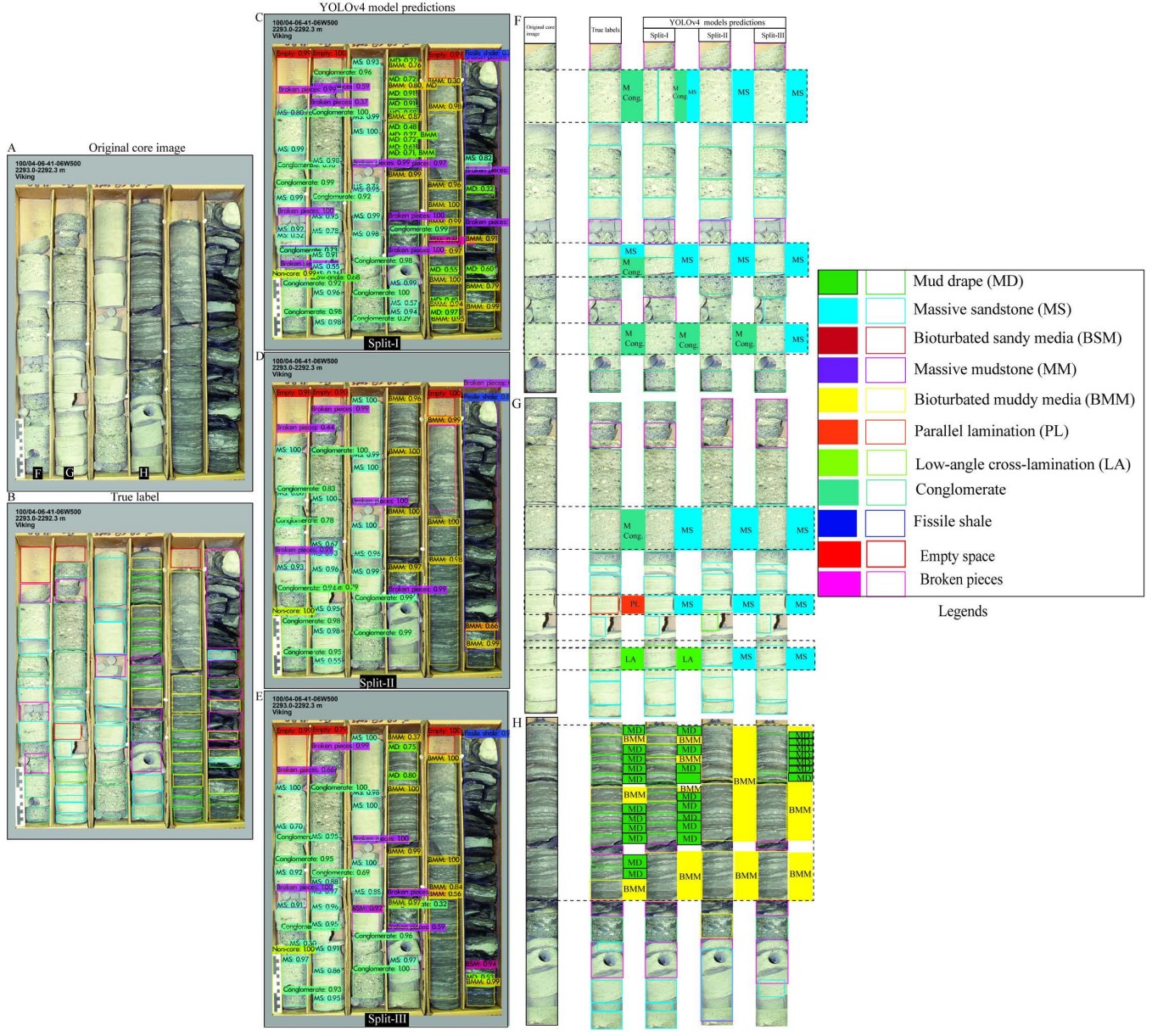

**Fig 8. Split-wise YOLOv4 predictions for diverse sedimentary structures in Dataset 1.** (A–B) Core image and annotations. (C–E) Detection results across splits. (F–H) Zoomed comparisons for detailed model evaluation.

## YOLOv4 versus Faster R-CNN

**Classification metrics.** YOLOv4-Darknet achieved consistently high values across key performance metrics, with precision, recall, and F1-score all reaching 0.95, 0.93, and 0.94, respectively (Table 4). This reflects the model's ability to both accurately and comprehensively detect sedimentary structures in core images. In contrast, Faster R-CNN exhibited slightly lower precision (0.9097) and a marked decline in recall (0.7496), which reduced its F1-score to 0.8017—indicating that it missed a notable proportion of true instances, particularly among more frequent classes such as massive sandstone and bioturbated muddy media. YOLOv4 also recorded a higher average IoU

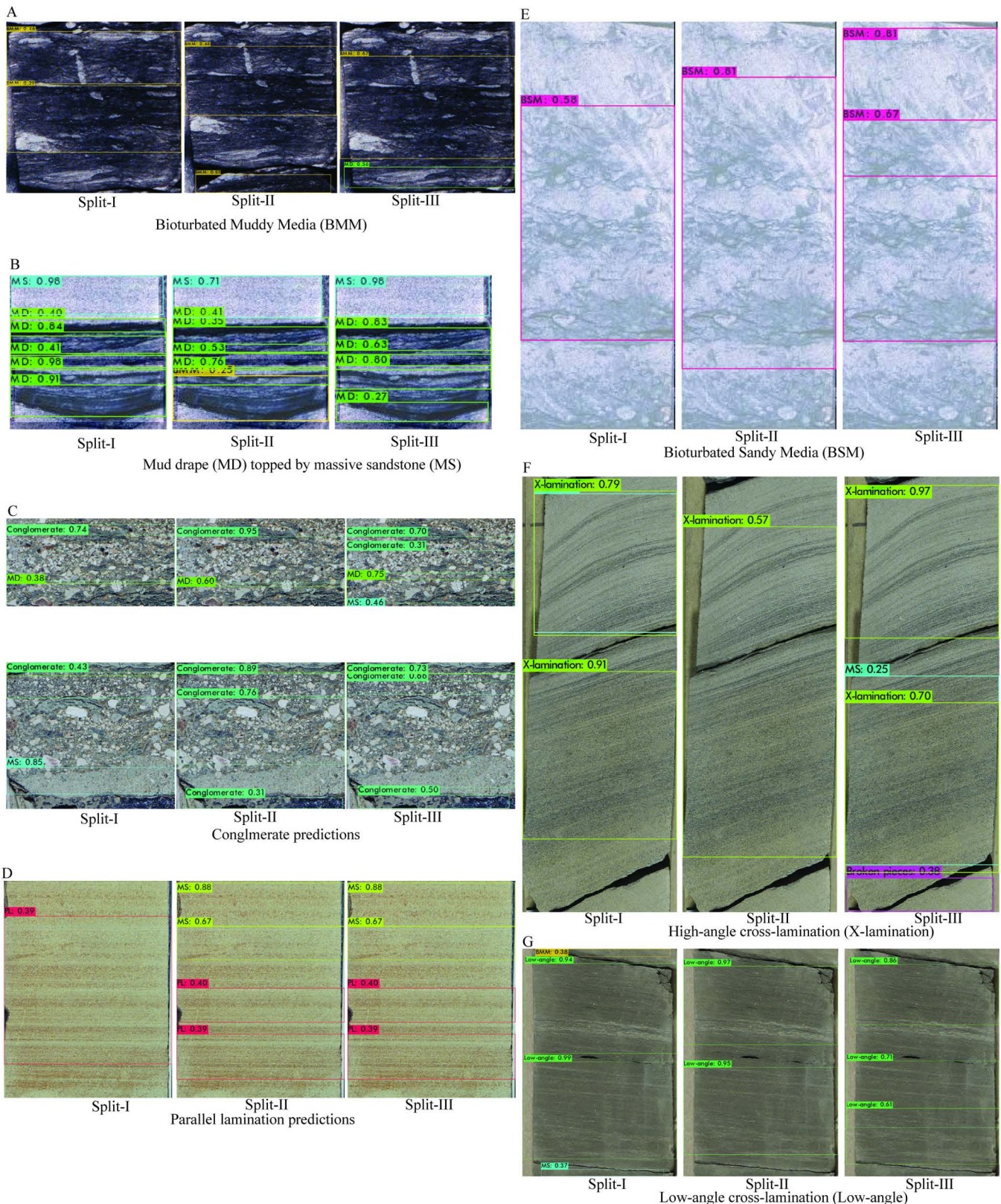

**Fig 9. Blind-test examples: (A) bioturbated muddy media; (B) massive sandstone with mud drape; (C) conglomerate; (D) parallel lamination; (E) bioturbated sandy media; (F) high-angle cross lamination; (G) low-angle cross lamination.** Color-coded boxes indicate model detections.

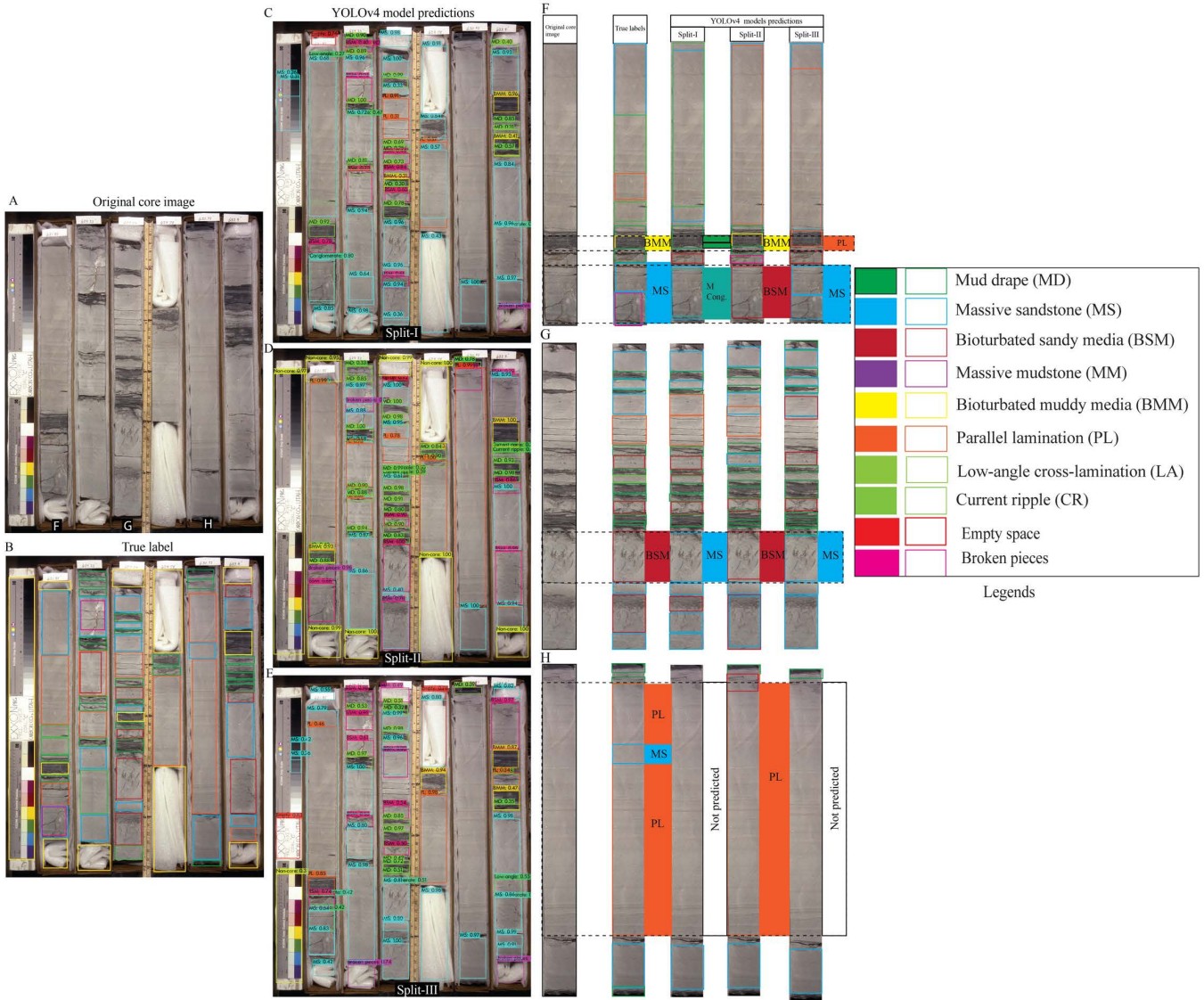

**Fig 10. YOLOv4 performance on Dataset 2 images.** (A–B) Original image and true labels. (C–E) Predictions for Splits I–III. (F–H) Regional comparison of predictions versus annotations.

(80.84% vs. 78.94%), demonstrating more precise spatial localization. However, Faster R-CNN outperformed in mean average precision (mAP), achieving 94.44% compared to 92.82% for YOLOv4. This suggests that while YOLOv4 excelled in detecting common structures, Faster R-CNN was more effective at classifying a wider range of sedimentary classes, including rarer types, albeit with lower recall. Training speed of YOLOv4 was five times faster than Faster R-CNN

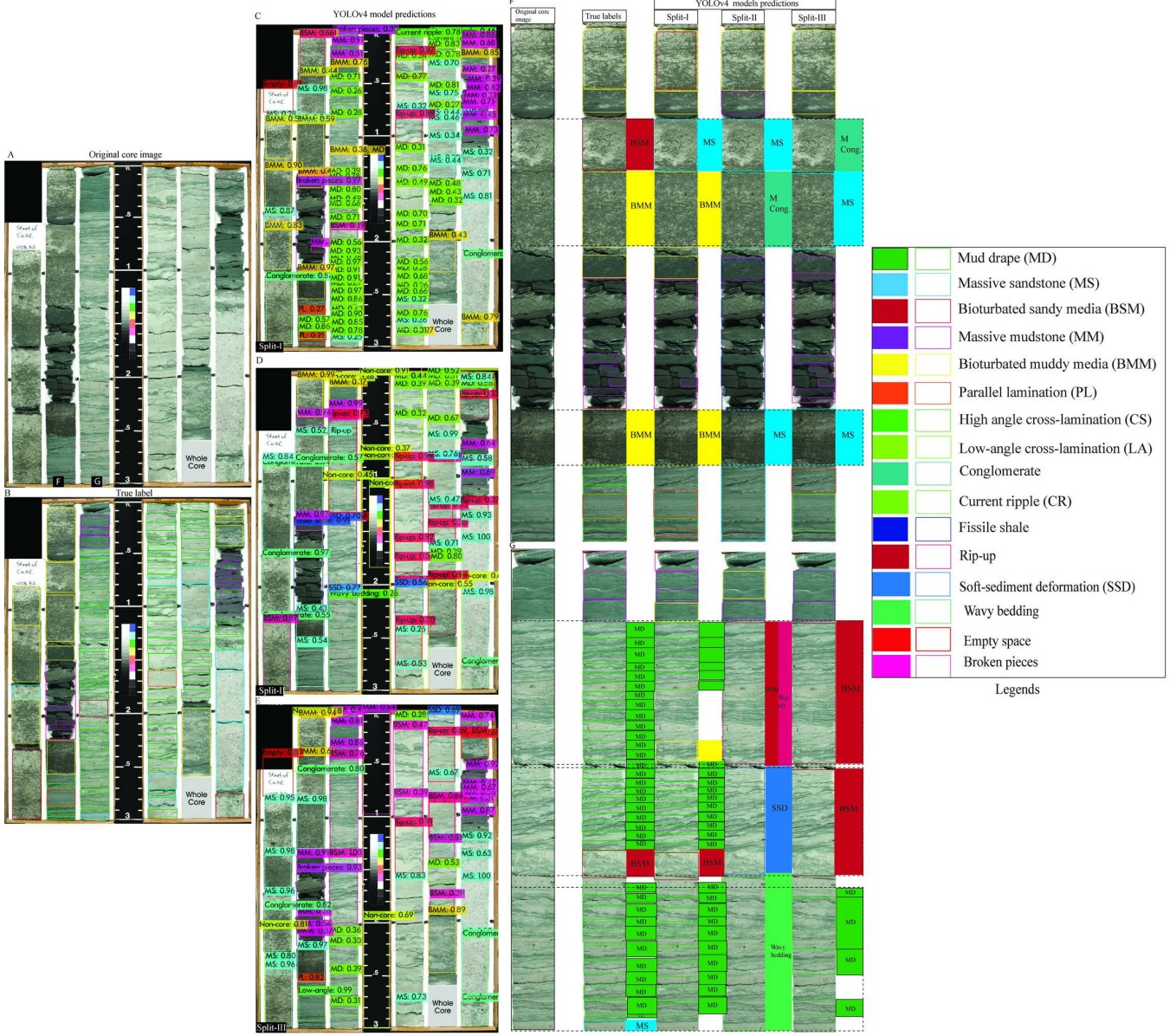

**Fig 11. YOLOv4 results on Dataset 3.** (A–B) Image and corresponding labels. (C–E) Predictions by split. (F–G) Column-wise comparison of predicted versus actual labels.

**Table 4. Comparative performance of YOLOv4-Darknet and Faster R-CNN-ResNet50 across key metrics: precision, recall, F1-score, Average IoU, mAP, and inference time.**

| Metrics | YOLOv4-Darknet | Faster R-CNN-ResNet50 |
|---|---|---|
| Precision | 0.95 | 0.9097 |
| Recall | 0.93 | 0.7496 |
| F1-score | 0.94 | 0.8017 |
| Average IoU | 80.84% | 78.94 |
| Mean average precision (mAP) | 92.82% | 94.44 |
| Inference time per image | 3.2 s | 2.5 s |

In addition to training duration, inference speed was assessed using 10 representative test images under identical hardware conditions. YOLOv4-Darknet required approximately 32.0 seconds (3.2 s per image), while Faster R-CNN completed the same set in 25.0 seconds (2.5 s per image). Although YOLOv4 is architecturally a one-stage detector optimized for real-time applications, its slightly slower inference in this case may stem from the computational demands of its Darknet-53 backbone and associated post-processing. Conversely, Faster R-CNN's use of ResNet-50-FPN and PyTorch's efficient tensor handling may have contributed to faster execution.

For classes with substantial representation in the dataset—such as mud drapes and massive sandstone—both YOLOv4 and Faster R-CNN achieve high precision, though Faster R-CNN often demonstrates a slight advantage, particularly in classes with complex textural or morphological nuances (Table 5). In underrepresented classes like concretions and scattered pebbles, Faster R-CNN generally exhibits superior precision, reflecting its strength in handling rare or intricate features. However, this trend is reversed in the case of broken pieces, where Faster R-CNN fails to detect any instances (0% precision), while YOLOv4 achieves a notably high precision of 98.71%.

### Test evaluation

Three core box images from Dataset 1 were employed to assess the comparative detection capabilities of YOLOv4 and Faster R-CNN (Fig 12). In the first test image, Faster R-CNN failed to identify several bioturbated sandy intervals that YOLOv4 correctly detected, although both models accurately recognized the remaining structures. In the second example, Faster R-CNN missed key features including mud drapes, low-angle cross stratification, and massive mudstone. In the third image, both models successfully predicted most occurrences of conglomerate, massive sandstone, mud drapes, and bioturbated muddy media. Nevertheless, misclassifications were noted—particularly confusion between conglomerate and

**Table 5. Class-wise average precision comparison between YOLOv4-Darknet and Faster R-CNN-ResNet50, including percentage representation in the dataset to contextualize model performance across sedimentary and non-rock classes.**

|  |  | Average precision (%) | |
|---|---|---|---|
|  | **Classes** | **YOLOv4-Darknet** | **Faster R-CNN-ResNet50** |
| Sedimentary structure | Mud drapes | 82.39 | 97.74 |
|  | Massive sandstone | 98.01 | 99.79 |
|  | Bioturbated muddy media | 96.51 | 99.97 |
|  | Massive mudstone | 96.95 | 99.85 |
|  | Bioturbated sandy media | 98.04 | 99.97 |
|  | Parallel lamination | 97.44 | 99.48 |
|  | Low-angle lamination | 98.42 | 99.61 |
|  | Massive conglomerate | 99.57 | 99.98 |
|  | Cross stratification | 98.75 | 99.95 |
|  | Current ripples | 84.06 | 98.78 |
|  | Fissile shale | 99.99 | 99.97 |
|  | Rip-up clasts | 98.93 | 99.96 |
|  | Scattered pebble | 25.6 | 99.73 |
|  | Concretions | 91.79 | 99.73 |
|  | Soft-sediment deformation | 100 | 100 |
|  | Wavy bedding | 100 | 100 |
| Non-rock | Broken pieces | 98.71 | 0 |
|  | Empty | 99.95 | 99.95 |
|  | Non-core | 98.45 | 99.98 |

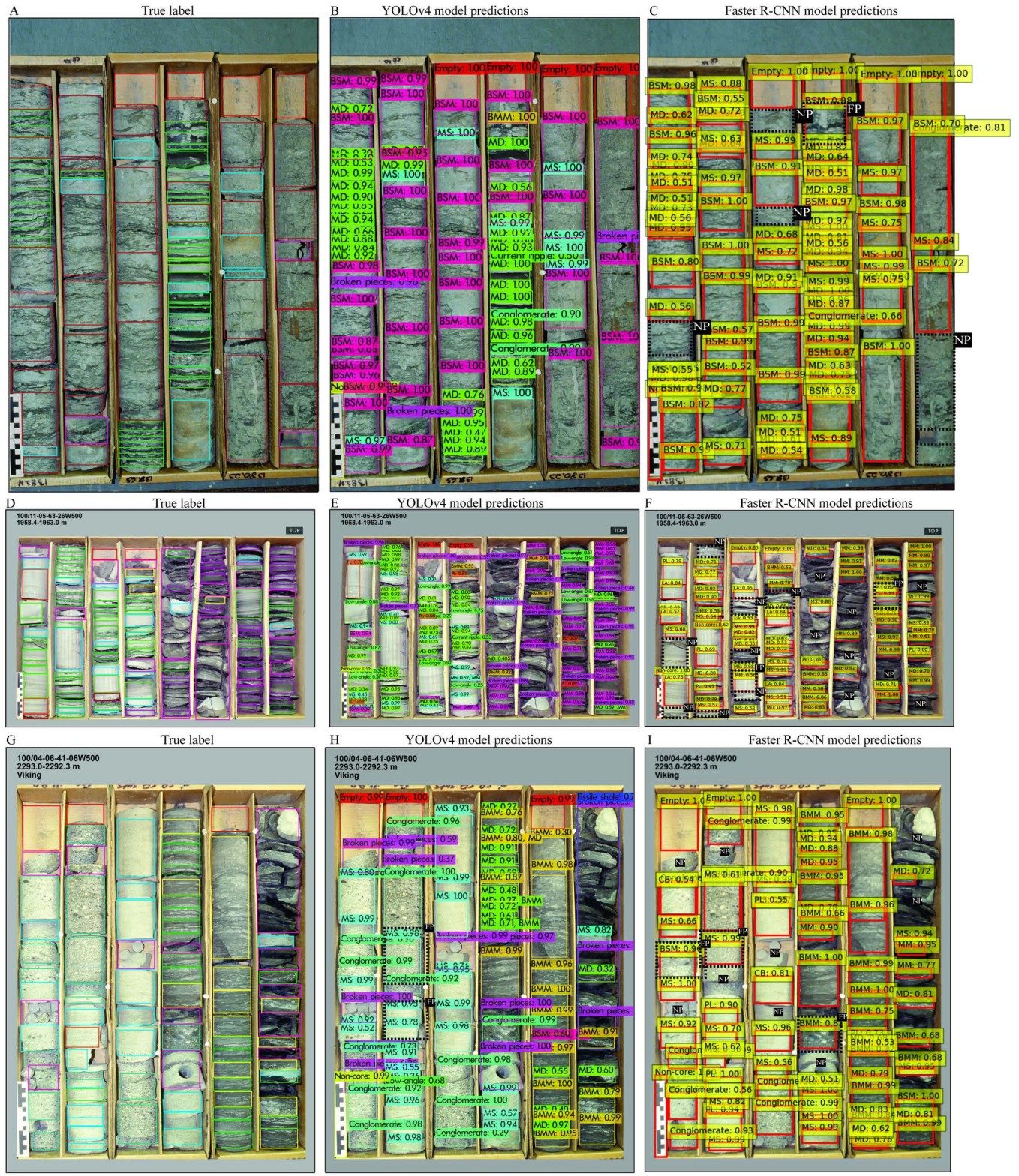

**Fig 12. Comparison of YOLOv4 and Faster R-CNN predictions for Dataset 1.** (A, D, G) True annotations. (B–C, E–F, H–I) Detection outputs from both models for three representative core images.

massive sandstone, and occasional incorrect labeling of conglomerate as bioturbated sandy or muddy media by Faster R-CNN. Overall, the majority of errors were observed in Faster R-CNN predictions.

Further evaluation using a representative image from Dataset 2 revealed similar limitations in Faster R-CNN's performance. Several mud drapes were missed, and there was significant confusion among parallel lamination, low-angle cross stratification, and massive sandstone. Additionally, bioturbated sandy media were frequently misclassified as massive sandstone or high-angle cross stratification. In contrast, an image from Dataset 3 demonstrated Faster R-CNN's relative strength in distinguishing between bioturbated sandy and muddy media—an area where YOLOv4 showed less accuracy. However, Faster R-CNN again failed to detect multiple instances of massive mudstone and mud drapes (Fig 13).

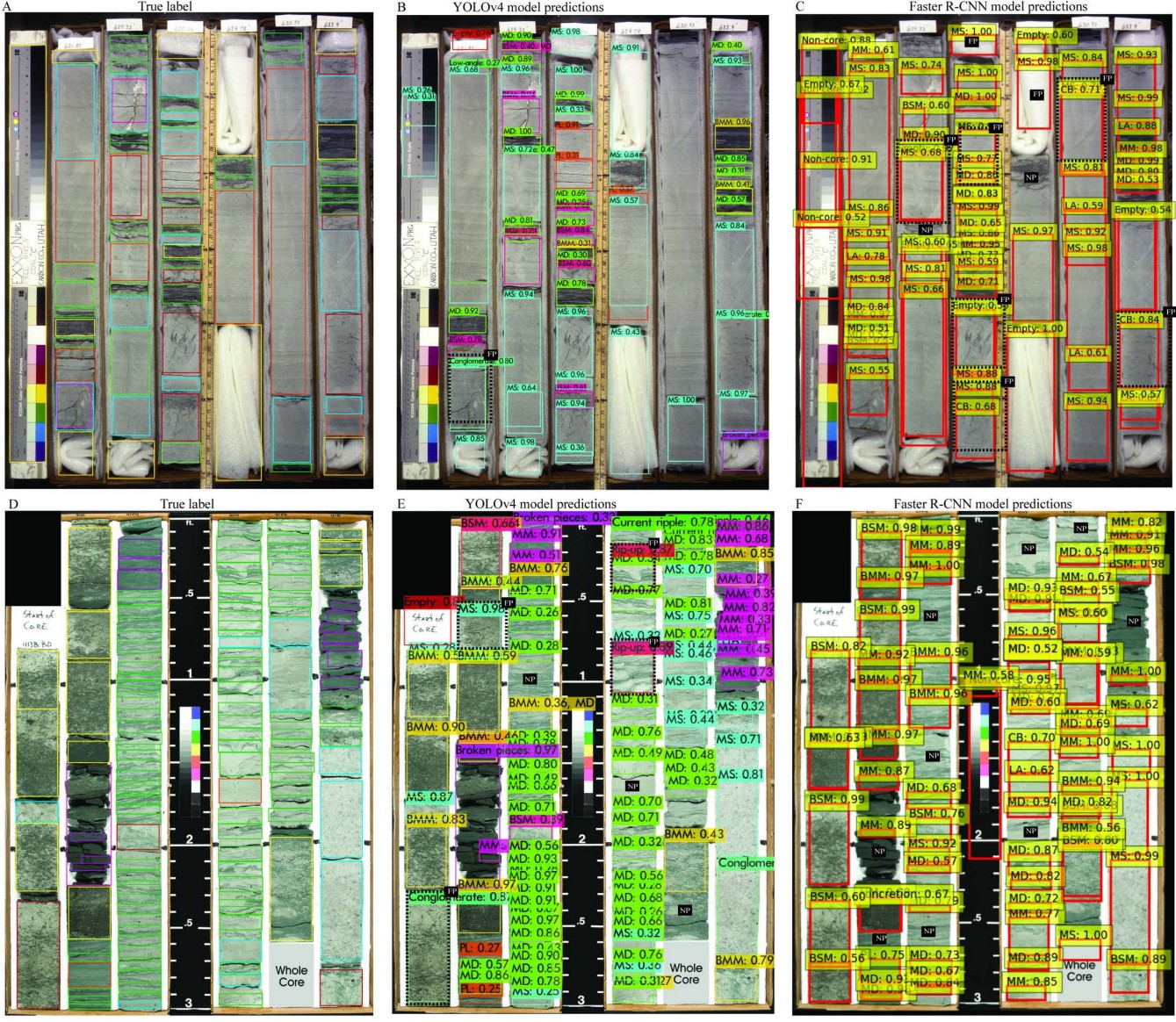

**Fig 13. Model predictions from YOLOv4 and Faster R-CNN for core images from Datasets 2 and 3.** (A, D) Ground truth. (B–C, E–F) Detection results illustrating model differences in classification accuracy.

## Discussion

Automating the detection of sedimentary structures can significantly improve the efficiency and consistency of core interpretation, especially given the diverse and often subtle visual expressions of these features. While prior efforts—such as Zhang et al. [37]—have demonstrated the potential of CNNs for classifying a limited set of sedimentary structures, our study represents the first large-scale application of object detection for this task. By leveraging YOLOv4 and Faster R-CNN architectures, we expanded the classification scope to 15 distinct sedimentary structures. A comparative summary (Table 6) illustrates advancements in model architecture, data volume (~134,500 bounding boxes), annotation granularity (bounding boxes vs. single-label classification), and validation strategies, establishing the present work as a foundational step toward scalable, interpretable sedimentological automation.

The results indicate that deep learning models can effectively identify common sedimentary structures, offering a reliable and efficient alternative to manual interpretation. YOLOv4, in particular, showed strong performance on prevalent classes such as massive sandstone and bioturbated muddy media (Table 2). However, mud drapes—despite comprising nearly a quarter of the dataset—yielded slightly lower precision scores (82.4%, 89.3%, and 84.7% across Splits I–III; Table 3), reflecting recurrent confusion with bioturbated sandy or muddy media. This misclassification likely stems from visual overlap, particularly in settings where bioturbated mud layers mimic the appearance of discontinuous and biogenically-disrupted mud drapes or resemble thin, massive mudstone beds (Figs 6g, 7f, 8h, 9a–9b, 11g).

Similar errors were observed between faint stratified structures (parallel and cross laminations) and massive sand-stone, especially in Split-III where prediction precision declined (Figs 7f–7h, 8g, 10g–10h). Rare classes such as soft-sediment deformation and siderite concretions were often assigned overly optimistic precision values, exceeding 90% in Split-I, despite accounting for less than 1% of bounding boxes. More balanced performance emerged in Splits II and III, likely due to increased data diversity from Datasets 2 and 3.

Misclassifications primarily occurred among classes with overlapping morphological traits. Discontinuous mud drapes within flaser bedding were often mistaken for bioturbated intervals (Figs 10f, 11g). Conglomerates were misclassified as massive sandstone when clast boundaries were indistinct—especially under suboptimal lighting or image quality (Fig 8). Although all classes were delineated using sedimentological criteria and expert-defined contextual rules, these outcomes highlight the persistent challenge of distinguishing transitional or visually similar features. Future improvements could involve higher-resolution imagery, integration of stratigraphic context, or development of texture-aware architectures to improve interpretive fidelity.

### Three splits of YOLOv4

The evaluation of YOLOv4 across three distinct training-validation-test splits provides insight into the model's robustness and consistency under varying data conditions (Table 2). In Splits I and II, precision, recall, and F1-score remain

**Table 6. Comparison between Zhang et al. [37] and this study in terms of methodology, dataset scale, model architecture, and evaluation metrics.**

| Study | Zhang et al. (2021) | This Study |
|---|---|---|
| Approach | CNN with Transfer Learning | Object Detection (YOLOv4, Faster R-CNN) |
| Dataset Size | 695 images (604 optimized) | 506 full core-images, ~40,000 annotation boxes (augmented to 134,500 boxes) |
| Classes | 3 (parallel bedding, wavy bedding, carbonaceous foliation) | 16 sedimentary structures + background |
| Validation Method | Random test-train split (70−30%) | 3 structured splits: Split-I, II, III |
| Performance | Accuracy ~91.11% (F1~86.96%) | YOLOv4: Precision/Recall ~94%, mAP 91%; Faster R-CNN: mAP 94.4%, lower recall |
| Output | Classification (label-only) | Object detection (bounding boxes, class labels) |

consistently high, suggesting the model performs effectively when trained on data with representative diversity and balanced class distributions. This indicates that YOLOv4 is well-suited to scenarios where training and validation datasets share similar sedimentological characteristics and class frequencies.

In contrast, Split-III shows a marked decline in performance, particularly in precision, pointing to the model's difficulty in generalizing when encountering less familiar or underrepresented features. The increased rate of false positives and missed detections in this split suggests that class distribution imbalances or dataset heterogeneity may have contributed to model underperformance. To clarify this, we refer readers to Supplementary S1–S3 Tables, which detail the number of bounding boxes for each of the 15 sedimentary structure classes across all splits. These tables confirm that all classes are represented in the test sets, thus eliminating class absence as a source of the observed decline in Split-III.

The model's reduced precision in Split-III was particularly evident in intervals characterized by visually or texturally similar features. Misclassifications occurred frequently between bioturbated muddy and sandy media, and between mud drapes and thin massive mudstone beds. These errors were more pronounced in Split-III, where training data may have contained fewer examples of such transitional or morphologically ambiguous structures. This pattern highlights a current limitation in object detection for sedimentological applications: distinguishing between subtly varying features with overlapping textural or structural characteristics. Addressing this issue may require integrating additional sedimentological context, such as stratigraphic position or high-resolution textures, into future models. The variability observed across splits reinforces the importance of comprehensive data preprocessing and augmentation to enhance model generalization.

## One-stage versus two-stages detections

YOLOv4 and Faster R-CNN represent distinct object detection paradigms—one-stage and two-stage architectures, respectively—each offering complementary strengths in sedimentary core analysis. YOLOv4 processes bounding box localization and classification in a single pass, enabling efficient training (24 hours) and, theoretically, faster inference, making it well-suited for high-throughput or real-time applications. However, its streamlined design may reduce precision when encountering rare or visually ambiguous features.

Faster R-CNN, by contrast, separates region proposal and classification steps, facilitating more detailed and focused analysis. This design contributes to improved detection of rare or morphologically complex sedimentary structures, as reflected in its higher mean average precision (94.44% vs. 92.82% for YOLOv4). Despite requiring longer training (~120 hours) than YOLOv4 (24 hours) and lower recall and F1-score metrics (Table 4), it proved effective in recognizing fine distinctions.

Contrary to expectations based on architectural complexity, empirical benchmarking on 10 test images showed that Faster R-CNN (2.5 s/image) outperformed YOLOv4 (3.2 s/image) in inference speed. This result likely stems from the computational demands of the YOLOv4-Darknet backbone and its post-processing, compared to the ResNet50-FPN backbone in Faster R-CNN and its implementation within the PyTorch framework. Notably, previous studies have also reported similar outcomes, where Faster R-CNN exhibited faster inference than YOLOv4-Darknet under certain conditions, particularly when using unoptimized or ported YOLOv4 implementations in PyTorch or TensorFlow [48,49]. These findings reinforce the importance of context-dependent benchmarking, where real-world deployment performance can diverge from theoretical assumptions due to framework-level optimizations and hardware compatibility.

In practical image tests (Figs 12–13), YOLOv4 produced more stable predictions for commonly occurring features such as massive sandstone, mud drapes, and bioturbated media. While Faster R-CNN occasionally excelled in distinguishing bioturbated sandy from muddy media in Dataset 3, it also exhibited more frequent misclassifications involving massive mudstone, low-angle cross stratification, and mud drapes. These results affirm that YOLOv4 is better suited for general feature coverage, whereas Faster R-CNN is advantageous in scenarios demanding precision for rare structures.

## Challenges related to labeling inconsistency

Establishing consistent and sedimentologically meaningful labeling criteria was essential for reliable model performance. Each class was defined based on expert interpretation and established sedimentary principles. For example, the massive sandstone class included both structureless and faintly laminated textures, while low-angle and high-angle cross stratification were distinguished using a protractor-based 10° threshold. An attempt to apply brightness thresholds via ImageJ software to differentiate sandy from muddy substrates was unsuccessful due to inconsistencies in lighting and imaging conditions.

We paid particular attention to abundant but commonly misclassified classes, such as mud drapes and bioturbated media (Fig 10f). These were differentiated using consistent sedimentological definitions: mud drapes were annotated as fine-grained, laterally continuous to semi-continuous layers within heterolithic or interbedded successions, while bioturbated media exhibited irregular disruption and burrow textures within sandy or muddy matrices. Expert visual assessment and domain knowledge underpinned these definitions, promoting uniform labeling across the dataset and mitigating the impact of visual overlap on model predictions.

Initial stages of the study involved simplified classification tasks using only two or four of the most prevalent classes. This allowed us to identify and correct labeling inconsistencies early. Partitioning the full, imbalanced dataset into smaller, balanced subsets enabled focused refinement—particularly for critical classes such as bioturbated sandy/muddy media, mud drapes, and massive layers. This staged approach improved both annotation quality and model interpretability before scaling to the full 15-class framework.

## Class imbalance

To evaluate the impact of class imbalance on detection performance, we reduced the number of structure classes from 18 to 12 by removing those with fewer than 250 bounding boxes—excluding fissile shale, which was retained due to its geological relevance. This adjustment led to the exclusion of approximately 1,000 bounding boxes and resulted in a 2.5% increase in mean average precision. Nonetheless, minor decreases in the precision of certain classes were observed, likely due to the exclusion of visually similar features, which may have introduced noise during classification.

Importantly, class imbalance in our dataset reflects natural sedimentological heterogeneity. Common structures such as mud drapes and bioturbated media are frequently observed in siliciclastic deltaic and deep-marine successions, whereas rare features like wavy bedding or soft-sediment deformation are tied to specific depositional regimes and are thus inherently underrepresented. To maintain geological authenticity, we did not use oversampling, which risks generating redundant morphologies and overfitting. Instead, we implemented class weighting during training to balance the contribution of rare and abundant classes. This approach preserved the integrity of the dataset while helping the model better learn from underrepresented structures.

Moreover, some rare classes, such as wavy bedding, are composed of recurring features from more common classes (e.g., stacked, curved mud drapes), allowing the model to generalize these complex forms from foundational patterns. The model's ability to correctly identify wavy bedding in blind tests (Fig 11g) supports the effectiveness of this strategy.

To further improve balance without compromising interpretive quality, we employed targeted data augmentation using naturalistic transformations (brightness and exposure adjustment, Gaussian blur, and strategic cropping). This yielded a ~30% boost in performance compared to models trained without augmentation. Annotation consistency was maintained using precise sedimentological criteria. For example, mud drapes were labeled as fine-grained, laterally continuous to semi-continuous features within heterolithic facies. Curved and closely spaced drapes were annotated as wavy bedding, while irregular patches within sandy contexts were classified as bioturbated sandy media. These systematic distinctions contributed to more accurate and geologically coherent model predictions.

## Model implications, limitations, and application potentiality

Automated tools for facies analysis remain limited in their ability to interpret complex sedimentary environments. This study provides a foundational framework by employing deep learning-based object detection to identify a wide range of sedimentary structures—many of which are common but underrepresented in traditional taxonomies. When combined with prior work on bioturbation intensity [20], this framework approximates early-stage facies interpretation. While the models perform well in detecting discrete features such as mud drapes, bioturbated media, and cross-laminated beds, they remain first-order identification tools.

Future development should expand this framework to include additional features—such as fractures, uncommon sedimentary textures, and detailed bioturbation patterns—to support more advanced interpretations. Full facies classification, which entails sequence stratigraphy and paleoenvironmental reconstruction, exceeds the present capabilities of the models. Integrating stratigraphic context, higher-resolution inputs, and morphometric descriptors will be key to advancing model interpretability. Emerging approaches, such as instance segmentation or transformer-based models, may offer the granularity required for this next step.

Equally important is enhancing input data quality. While this study utilized static 2D imagery, many core images were captured with partial rotation, suggesting the feasibility of multi-angle or pseudo-3D imaging. Incorporating photogrammetric workflows could enable better morphological characterization and improve model understanding of spatial patterns.

Finally, while annotations were prepared with expert oversight, the model predictions were validated only on select blind test cases. We recommend future efforts include comprehensive expert review of outputs and formal inter-annotator agreement protocols to strengthen reproducibility and interpretive reliability.

## Conclusions

This study validates the application of deep learning—specifically YOLOv4 and Faster R-CNN—for automating the detection of sedimentary structures in core box imagery. Using object detection frameworks on a diverse siliciclastic dataset, we addressed challenges associated with manual interpretation and examined the trade-offs between speed, precision, and class sensitivity across models.

YOLOv4 exhibited strong performance in detecting abundant classes such as massive sandstone, bioturbated muddy media, and parallel lamination, with high precision, recall, and F1-scores in Splits I and II. However, its accuracy declined for visually complex or rare classes—such as mud drapes and soft-sediment deformation—particularly in Split III. Faster R-CNN outperformed in mean average precision (94.44%) and in identifying underrepresented structures but showed lower recall and inconsistent predictions for common features.

Unexpectedly, inference benchmarking revealed that Faster R-CNN achieved shorter prediction times (2.5 s/image) compared to YOLOv4 (3.2 s/image), despite its two-stage architecture. This inversion of expected performance likely reflects the increased computational demands of the YOLOv4-Darknet backbone and post-processing load, compared to the more optimized ResNet50-FPN implementation in Faster R-CNN under the PyTorch framework.

Overall, YOLOv4 provides a balanced and scalable approach suitable for high-throughput sedimentary analysis, while Faster R-CNN is better suited for cases requiring enhanced detection of rare or morphologically subtle structures. Used individually or in tandem, both models offer significant advances in the efficiency, consistency, and resolution of core interpretation workflows.

Future directions include diversifying training data, integrating stratigraphic and morphometric context, and exploring advanced architectures such as instance segmentation or transformer-based models to expand the interpretive capabilities of AI-driven sedimentology.

## Supporting information

**Supplementary Table S1. Distribution of bounding boxes for each sedimentary structure across training and test sets in Split-I. This table highlights class representation balance to ensure effective model training and evaluation.** (XLSX)

**Supplementary Table S2. Distribution of bounding boxes for each sedimentary structure across training and test sets in Split-II. The table illustrates the consistency of data partitioning for reproducibility.**
(XLSX)

**Supplementary Table S3. Distribution of bounding boxes for each sedimentary structure across training and test sets in Split-III. It confirms that all classes are represented, supporting fair performance evaluation despite observed precision drops.**
(XLSX)

**Supplementary Table S4. Definitions and formulations of classification metrics used in model evaluation. Includes precision, recall, F1-score, Average IoU, and mean average precision (mAP) to facilitate interpretation of model performance across different detection tasks.**
(CSV)

## Acknowledgments

The authors would like to acknowledge the support received from Saudi Data and AI Authority (SDAIA) and King Fahd University of Petroleum and Minerals (KFUPM) under SDAIA-KFUPM Joint Research Center for Artificial Intelligence Grant no. JRC-AI-RG-03. They would like to thank Dr. Ardiansyah Koeshidayatullah for insightful discussions during the early stages of this research.

## Author contributions

**Conceptualization:** Ammar Abdlmutalib, Korhan Ayranci, Umair Bin Waheed.

**Data curation:** Ammar Abdlmutalib, Korhan Ayranci, James A. MacEachern.

**Formal analysis:** Ammar Abdlmutalib.

**Funding acquisition:** Korhan Ayranci, Umair Bin Waheed.

**Investigation:** Ammar Abdlmutalib, Umair Bin Waheed, Hamad D. Alhajri.

**Methodology:** Ammar Abdlmutalib, Korhan Ayranci, Umair Bin Waheed, Hamad D. Alhajri, James A. MacEachern, Mohammed N. Al-Khabbaz.

**Project administration:** Korhan Ayranci.

**Resources:** Korhan Ayranci, Umair Bin Waheed.

**Software:** Ammar Abdlmutalib, Umair Bin Waheed, Hamad D. Alhajri, Mohammed N. Al-Khabbaz.

**Supervision:** Korhan Ayranci, Umair Bin Waheed, James A. MacEachern.

**Validation:** Ammar Abdlmutalib, Korhan Ayranci, Umair Bin Waheed, Hamad D. Alhajri, James A. MacEachern.

**Visualization:** Ammar Abdlmutalib, Hamad D. Alhajri.

**Writing – original draft:** Ammar Abdlmutalib, Korhan Ayranci, Umair Bin Waheed, Hamad D. Alhajri, James A. MacEachern.

**Writing – review & editing:** Ammar Abdlmutalib, Korhan Ayranci, Umair Bin Waheed.

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
