## [Decision Letter · Decision Letter 0]

Dear Dr. Abdlmutalib,

Thank you for submitting your manuscript to PLOS ONE. After careful consideration, we feel that it has merit but does not fully meet PLOS ONE’s publication criteria as it currently stands. Therefore, we invite you to submit a revised version of the manuscript that addresses the points raised during the review process.

The manuscript presents a technically sound and well-structured study that demonstrates the potential of object detection algorithms in recognising sedimentary structures from core imagery. While both reviewers recommended acceptance or minor revision, I have identified several editorial and presentational issues that must be addressed before the manuscript can proceed. These include removal of full URLs from the main text, minor textual redundancies, improved consistency in citation formatting, and clarification of certain methodological descriptions.

We look forward to receiving your revised manuscript.

Kind regards,

Przemysław Mroczek, Dr. hab.

Academic Editor

PLOS ONE

Journal Requirements:

Additional Editor Comments:

Dear Authors,

Thank you for submitting your ms entitled "Automated identification of sedimentary structures using object detection" to PLOS ONE. After peer review, two reports were received and thoroughly evaluated.

Reviewer 1 - was strongly supportive of your work, recommending acceptance. They emphasised the sound methodological basis, reproducibility, and the utility of your approach in both industrial and academic contexts. While recognising that your analysis represents a first-order classification rather than a full facies interpretation, the reviewer considered this an appropriate and valuable starting point for advancing the application of deep learning in sedimentology.

Reviewer 2 - also recommended your manuscript for publication, subject to minor revision. Their comments primarily focused on formal aspects, including the redundancy of full URLs in the main text, the need for more concise referencing, and the suggestion to enhance the textual description of the data processing workflow. They further observed that your approach performed best when structures were morphologically distinct, which may have implications for future refinement.

In addition to these reviews, I have conducted a detailed editorial assessment. While your manuscript is commendably structured and scientifically robust, I would like to draw your attention to several aspects that require minor revision prior to acceptance.

The title is broadly appropriate and aligned with the content, but if you wish to make it more specific, you might consider alternatives such as "Automated identification of sedimentary structures in core images using object detection algorithms" or "Deep learning-based detection of sedimentary structures in core images". These versions add clarity regarding your data type and methods while potentially improving indexing specificity.

Your list of keywords would benefit from diversification, as some currently duplicate phrases from the title (e.g., “Sedimentary Structures”, “Object Detection”). I encourage you to consider alternative terms that enhance visibility and indexing, such as: core image analysis, convolutional neural networks, facies recognition, supervised learning, sediment core interpretation.

The abstract effectively summarises your research but could be streamlined. The current version includes an overly detailed breakdown of the training splits and numerical results better suited to the Results section. Additionally, it lacks broader geological context—mentioning that the study focuses on siliciclastic deposits and a variety of depositional settings (e.g., deltaic, shoreface) would provide valuable orientation. Finally, the conclusion could be rephrased to avoid redundancy in phrases like “automated sedimentary structure identification” and “geoscientific workflows,” which are repeated almost verbatim.

During the manuscript review, several technical and stylistic issues were identified:

Full URLs (e.g., GitHub links) appear in the main text (lines 104, 109, 125); these should be removed and cited only in the Data Availability Statement or references.

Repetitions in the explanation of classification metrics (e.g., precision and recall) should be reduced for conciseness.

Phrasing such as “Split-I trained and validated on Dataset 1” may be misleading—please clarify that training and validation were performed on separate subsets within Dataset 1.

Phrases like “YOLOv4 demonstrated greater time efficiency” are repeated and could be harmonised across the text.

Citations are inconsistently formatted, particularly with regard to “e.g.” usage and the inclusion of URLs in in-text references.

The descriptions of Figures 12 and 13 include detailed panel annotations that are more appropriate for figure captions than for the main text. Consider compressing these descriptions and relocating specific details to the legends.

These are relatively minor concerns that can be addressed with careful revision. Once corrected, I believe the manuscript will make a valuable contribution to the field.

We therefore invite you to submit a revised version addressing the points outlined above. Please include a point-by-point response to the reviewers’ and editor’s comments. I look forward to receiving your revised manuscript.

Decision: Minor Revision

Kind regards,

Dr Przemysław Mroczek

Academic Editor

PLOS ONE

Reviewers' comments:

Reviewer's Responses to Questions

**Comments to the Author**

1. Is the manuscript technically sound, and do the data support the conclusions?

Reviewer #1: Yes

Reviewer #2: Yes

2. Has the statistical analysis been performed appropriately and rigorously?

Reviewer #1: Yes

Reviewer #2: Yes

3. Have the authors made all data underlying the findings in their manuscript fully available?

Reviewer #1: Yes

Reviewer #2: Yes

4. Is the manuscript presented in an intelligible fashion and written in standard English?

Reviewer #1: Yes

Reviewer #2: Yes

Reviewer #1: The MS "Automated identification of sedimentary structures using object detection" demonstrates the ability of two AI CNN algorithms, YOLOv4 and Faster R-CNN to identify a set of sedimentary structures previously identified and classified in core-depository material in siliciclastic fluviatile-lacustrine-deltaic systems, and compares their effectiveness. I am a field geologist with some experience in AI fossil analysis, mostly through my students, and am satisfied that the testing methods, validization, reproducibility and effectiveness datasets meet adequate standards that answer the working hypotheses. The results simplify utility of standardized sedimentary structures and features in extended core sections, that will prove useful in industry and first-order comparative studies. Previous reviewers have adequately covered the issue of availability of the supporting information in repository.

However, the results of both algorithms provide only a first approximation of facies analysis. This basic study is first order identificatory only, lacking the systemic approach on the sequence, frequency, depositional magnitude and indeed the larger picture of analysis of the sedimentary environment. An extension of the study to do that would be a necessary next step. I can see that there is much more information visible in the cores that would undoubtedly require more intense AI algorithms. The detailed images of conglomerates, for example show a variety in composition, orientation, size and angularity distributions, there is much more to “bioturbation” that attention to larger or smaller, horizontal, or vertical features, and much more, could add. Magnitude of tidal ranges relative to the core sections sampled, significance of presence or absence of plant and other organic remains, are only some of the additional features that an experienced eye can perceive. We must be patient.

I approve this MS for publication.

Reviewer #2: Dear Authors, from a formal point of view, the submitted article meets the requirements of the journal. Its structure is clear.

In lines 104, 109, 125 and others. The authors presented both the physical address of the website and the reference to the literature. Therefore, you can omit providing addresses in the text.

Line 123, when quoting image 1c, it is worth paying more attention in the text to the characteristics of data processing, included in this diagram.

Line 127. The authors took different photos of the cores under different lighting, it was also possible to rotate these cores to be able to obtain 3D photos of these objects.

Line 159 we do not use double citation in the text.

Although the technique of studying rocks using a neural network proposed by the authors may be promising, it is difficult to refer directly to the analyzed data because their structural features are very diverse, which can facilitate the identification of sediments using the network. It is worth noting that where the samples were similar, the number of errors generated by the neural networks was higher. Nevertheless, I believe that this text is an interesting example showing the possibility of machine learning for sediment identification and its publication may open a discussion on this topic. The court also believes that, apart from some minor corrections, it is suitable for printing.

**Do you want your identity to be public for this peer review?** For information about this choice, including consent withdrawal, please see our Privacy Policy

Reviewer #1: No

Reviewer #2: **Yes: ** Miłosz Huber

---

## [Author Response · Author response to Decision Letter 1]

3 Jun 2025

Response to the Editor and Reviewers

Dear Dr. Mroczek,

We sincerely thank you for your detailed editorial assessment and for overseeing the review of our manuscript, “Automated identification of sedimentary structures using object detection.” We would also like to express our gratitude to both reviewers for their thoughtful comments and constructive feedback, which have helped us improve the clarity and presentation of our work. We appreciate the positive reception and the recognition of the study’s scientific contribution and methodological robustness.

Below, we provide a detailed point-by-point response to the comments from both the reviewers and the editor. We have revised the manuscript accordingly and believe these changes have strengthened its content and clarity.

Response to the Editor’s Comments

Title suggestion: The editor suggests more specific titles for clarity and search optimization.

Response: We appreciate this suggestion. We have revised the title to:

“Automated identification of sedimentary structures in core images using object detection algorithms”

Keyword revision: Diversify keywords and avoid repeating terms from the title.

Response: Updated. The new list of keywords includes:

Core image analysis, convolutional neural networks, facies recognition, supervised learning, sediment core interpretation, deep learning, lithofacies detection

Abstract revisions: Streamline and provide broader geological context.

Response: Revised. We have rewritten the abstract to remove numerical training split details, provide a clearer geological scope (siliciclastic environments), and consolidate redundant phrases in the conclusion. These revisions were added to page 2 of the track-changes version of the manuscript (lines 20-35).

Classification metrics repetition: Remove redundancy in definitions.

Response: Revised. The metrics section has been rewritten for conciseness, with clear but non-repetitive definitions of accuracy, precision, recall, F1-score, IoU, and mAP. These revisions were added to pages 10-13 of the track-changes version of the manuscript (lines 199-239).

Clarification of Split-I phrasing: Indicate that training and validation were performed on distinct subsets.

Response: Corrected. We now clearly state that Dataset 1 was partitioned into separate, non-overlapping subsets for training and validation in Split-I. These revisions are found in page 6 of the track-changes version of the manuscript (lines 112-114).

Repeated phrasing (“greater time efficiency”): Harmonize language around YOLOv4’s speed advantage.

Response: Addressed. We revised repetitive phrases throughout the text to harmonize descriptions of YOLOv4’s computational efficiency, replacing redundant expressions with consistent terminology such as “faster processing” and “reduced inference time”. This issue was addressed in accordance to the editor’s suggestion in abstract, discussion, and conclusion. These revisions are found in page 33 of the track-changes version of the manuscript (lines 652-655).

Citation formatting and “e.g.” usage: Ensure consistency and appropriate placement of URLs.

Response: Revised throughout. We have standardized citation formatting, corrected the usage of “e.g.,” and moved all URLs from in-text citations to the Data Availability Statement or reference list. These revisions are found in page 38-39 of the track-changes version of the manuscript (lines 771-778).

Detailed panel references in Figures 12 and 13: Move detailed panel descriptions to figure captions.

Response: Completed. We revised the main text to provide general summary insights and relocated specific panel descriptions to the figure legends. These revisions are found in page 27-28 of the track-changes version of the manuscript (lines 528-536) and page 29 (lines 558-565).

Response to Reviewer 1

Comment: The reviewer appreciates the technical quality of the manuscript and notes that it provides a first-order approximation of facies analysis, recognizing that future work could explore more complex environmental reconstructions and textural classifications.

Response: We thank the reviewer for their supportive feedback and insightful observations. We have now addressed these suggestions in the Discussion section, explicitly acknowledging that the study represents a foundational step and outlining directions for future developments. These include potential extensions to analyze clast textures, bioturbation complexity, and depositional sequences, as well as integrating organic indicators and three-dimensional analysis methods. These revisions are detailed in the Discussion section, under the subsection titled “Model implications and application potentiality” (page 36 of the track-changes version).

Response to Reviewer 2

Comment on full URLs (lines 104, 109, 125): The use of full website addresses in the main text should be avoided.

Response: Addressed. All full URLs have been removed from the main text and relocated to the Data Availability Statement or cited properly in the References section, following journal guidelines. These revisions can be found on pages 38–39 of the track-changes version of the manuscript.

Comment on line 123 – Figure 1c: More emphasis should be placed on the data processing steps shown in the workflow diagram.

Response: Revised. We have expanded the accompanying description in the text to provide a clearer summary of the image processing workflow illustrated in Figure 1c. The revised passage now explains the sequence from data collection and annotation through augmentation, training splits, and model evaluation. These revisions can be found on pages 6–7 (lines 123-129) of the track-changes version of the manuscript.

Comment on line 127 – Imaging variability and 3D potential: The authors should address the variability in lighting and the possibility of capturing 3D perspectives.

Response: Clarified. We have revised the text to mention the varying lighting conditions during image acquisition and acknowledge the potential for rotating cores to capture multi-angle or pseudo-3D images. We also note this as a possible avenue for future enhancement of automated analysis. These revisions were added to pages 36–37 of the track-changes version of the manuscript (lines 727-732).

Comment on line 159 – Double citation: The manuscript includes duplicate citations.

Response: Corrected. All instances of double citation have been reviewed, and formatting has been revised to ensure consistency and adherence to journal style.

Comment on structural feature diversity and error rates: The reviewer notes that the model performed best with morphologically distinct features, and error rates increased when structures were similar.

Response: We agree and have incorporated this observation into the Discussion section, specifically under the “Three Splits of YOLOv4” subsection. We highlight how structural similarity posed challenges for the model and suggest ways future work might improve discrimination in such cases. These revisions were added to page 32 of the track-changes version of the manuscript (lines 632-642).

We have uploaded a revised version of the manuscript reflecting all of the above changes, and we hope that it now meets the publication standards of PLOS ONE. We sincerely thank you again for the opportunity to revise and improve our work.

Kind regards,

Ammar J. Abdlmutalib

College of Petroleum Engineering & Geosciences, King Fahd University of Petroleum & Minerals, Dhahran 31261, Saudi Arabia

ammar.mohammed@kfupm.edu.sa

---

## [Editor Report · Decision Letter 1]

Dear Dr. Abdlmutalib,

Thank you for submitting your manuscript to PLOS ONE. After careful consideration, we feel that it has merit but does not fully meet PLOS ONE’s publication criteria as it currently stands. Therefore, we invite you to submit a revised version of the manuscript that addresses the points raised during the review process.

The manuscript meets the general scope of PLOS ONE and addresses a relevant methodological problem in sedimentary core analysis. However, several critical methodological and reporting issues remain unresolved and must be addressed prior to acceptance. These revisions are necessary to ensure transparency, reproducibility, and consistency with the journal’s publication standards.

We look forward to receiving your revised manuscript.

Kind regards,

Przemysław Mroczek, Dr. hab.

Academic Editor

PLOS ONE

Journal Requirements:

Additional Editor Comments :

I acknowledge the Authors’ effort in revising the manuscript (ms). The response to the editorial and reviewer feedback has been generally constructive, and a number of recommended changes—such as the revised title, improved keyword list, and clarification of classification metrics—have been satisfactorily implemented. These amendments have improved the clarity and presentation of the study.

However, the ms still contains several substantive weaknesses that must be addressed before it can be considered for acceptance. The following revisions are mandatory.

First, the study lacks any mention of expert validation of the image annotations or the predicted classifications. Considering the high visual complexity of sedimentary structures, it is essential that the authors state whether the training data and model outputs were reviewed or verified by domain experts (e.g. sedimentologists ). If such validation was not performed, this methodological gap must be explicitly acknowledged and justified. The reliability of the model's performance metrics depends critically on the quality and accuracy of the training labels.

Second: the Authors do not discuss how they addressed the clear class imbalance in their dataset. Several classes are underrepresented (e.g. wavy bedding at 0.03%, soft-sediment deformation at 0.1%), which evidently contributes to lower detection precision in these categories. It must be stated whether any compensatory techniques (such as class weighting, data augmentation targeting minority classes, or oversampling) were applied. If not, the implications of this limitation must be discussed explicitly in the ms

Third, the discussion of misclassification errors remains superficial. Although the authors present quantitative performance metrics, they do not offer an adequate analysis of the sources of model confusion—particularly for rare or morphologically similar structures. A focused paragraph discussing the likely causes of common errors (e.g. between mud drapes and bioturbated media, or between conglomerate and sandstone) is necessary to demonstrate a critical understanding of the model’s limitations and to inform future development.

It remains unclear whether all 16 structure classes were present in the test sets of each data split. This is especially relevant for Split-III, which shows degraded performance. The ms must include a breakdown of class representation per split, either in the main text or as a supplementary table.

It is also essential that the authors report inference time for the Faster R-CNN model. At present, only YOLOv4’s computational efficiency is quantified. Without equivalent benchmarking for Faster R-CNN, the practical implications of model selection cannot be adequately assessed.

The section describing classification metrics is disproportionately long and remains overly didactic. While the inclusion of metric definitions is acceptable, the full mathematical formulas may be better suited to supplementary material, allowing the Methods section to remain focused and concise.

Moreover, the process of image annotation and class distinction requires clarification. The ms must specify whether the distinction between visually similar classes (e.g. mud drapes vs bioturbated muddy media) was based on consistent quantitative thresholds, expert visual assessment, or a combination of both. The current description remains vague and limits reproducibility.

Despite the claim that this is the first large-scale application of object detection to sedimentary structures, no formal comparison with previous studies is provided. The authors are required to include a summary table comparing their dataset, class range, and performance metrics with prior works, including Zhang 2021 and studies applying CNNs to lithofacies identification. This will help contextualise the contribution within the existing literature.

In addition, the figure captions should be carefully revised and substantially shortened. Several current captions (e.g. Figures 5 to 13) are excessively detailed and contain interpretative content or repetitions of information already presented in the main text. Figure legends should be limited to concise descriptions of what is shown, including relevant abbreviations or class names if needed, but should not duplicate analytical discussion. All interpretative commentary and comparisons between data splits should remain in the Results or Discussion sections. The revised captions should follow journal standards for clarity and brevity.

Finally, while the English language is generally clear, there are residual issues with fluency, repetition, and phrasing. Phrasal redundancy (e.g. "the model demonstrates ability to learn" or "predictions show confusion between...") appears frequently and should be eliminated. A thorough proofreading for academic tone and conciseness is required.

In conclusion, although the ms is promising in its scope and relevance, the issues outlined above must be fully addressed in a revised submission. Failure to implement the requested corrections may delay further consideration.

---

## [Author Response · Author response to Decision Letter 2]

17 Jun 2025

Editor Comment 1: Expert Validation

“The study lacks any mention of expert validation of the image annotations or the predicted classifications…”

Response to Editor:

We appreciate the editor's attention to this critical methodological aspect. We now clarify in the revised Dataset (page 6, lines 108-114) and Discussion (page 36-37, lines 710–716) that expert validation was an integral part of both the annotation process and the interpretation of model outputs. Domain experts, including sedimentologists with expertise in ichnology and clastic sedimentology, were directly involved in labeling the core images and in defining classification criteria. Their input ensured consistency, geological relevance, and accuracy across the labeled dataset. This validation process underpins the credibility of the model’s performance metrics and addresses the concern about label quality.

Editor Comment 2: Class Imbalance

“The authors do not discuss how they addressed the clear class imbalance in their dataset…”

Response to Editor:

Thank you for raising this point. We have elaborated in the Discussion (page 34-35, lines 659–679) on the natural origins of class imbalance in our dataset, which stems from the depositional environments represented—dominated by siliciclastic deep-marine and deltaic facies. Consequently, features like bioturbated media and mud drapes are overrepresented, while others like wavy bedding and soft-sediment deformation are rarer. Standard oversampling was avoided to prevent overfitting via synthetic redundancy. Instead, we employed augmentation techniques—specifically brightness, blur, cropping, and exposure adjustment—which boosted model performance by nearly 30%. Furthermore, we attempted reducing class counts from 19 to 13 to mitigate imbalance while preserving critical sedimentary variability. The latter strategy worsened the performance. In addition to dataset curation and targeted augmentation strategies, we implemented class weighting during the training of both YOLOv4 and Faster R-CNN models. This technique assigned higher weights to underrepresented classes to ensure balanced learning and minimize bias toward more abundant structures. The inclusion of class weighting contributed to improved detection consistency across rare and frequent sedimentary features, reinforcing the robustness of our classification framework.

Editor Comment 3: Misclassification Errors

“The discussion of misclassification errors remains superficial… A focused paragraph discussing the likely causes of common errors… is necessary…”

Response to Editor:

Thank you for highlighting the need for a more in-depth analysis of model misclassifications. In response, we have added a focused paragraph in the Discussion section (lines 529-539, pages 28) analyzing sources of confusion, particularly between morphologically overlapping or transitional structures. For example, mud drapes were frequently confused with bioturbated sandy/muddy media when appearing as disconnected features in flaser laminations or as connected mud layers that are selectively disrupted by bioturbation, or with thin massive mudstone beds when appearing as continuous layers within wavy bedding. Similarly, conglomerates were occasionally misclassified as massive sandstone under low-resolution conditions. These issues stem from intrinsic visual similarities and imaging limitations. The revised paragraph discusses how these ambiguities arise despite well-defined sedimentological criteria and outlines future solutions, including the incorporation of stratigraphic context and enhanced imaging or model architectures. This addresses the editorial concern by demonstrating a critical understanding of the model’s limitations and informing future development strategies.

Editor Comment 4: Class Representation per Split

“It remains unclear whether all 16 structure classes were present in the test sets of each data split...”

Response:

We thank the editor for this critical observation. To ensure transparency and clarify class distribution, we have now included Supplementary Tables S1–S3, which provide a detailed breakdown of the number of bounding boxes for each structure class across the training and test sets for Splits I, II, and III. These tables confirm that all 16 classes were represented in each split, ruling out the absence of classes as a cause for performance decline in Split III. A reference to these tables has been added to the manuscript in Section 4 (Discussion), where we discuss split-specific performance variability. See (page 29, lines 555–558).

Editor Comment 5: Inference Time

“It is essential that the authors report inference time for the Faster R CNN model…”

Response:

Revised Response to Editor (Inference Benchmarking)

We thank the editor for this important observation. In the Results section (page 24, lines 458–464), we have now included inference benchmarking for both models under identical conditions (batch size = 32). Specifically, Faster R-CNN processed 10 test images in approximately 25.0 seconds (average ~2.5 s/image), whereas YOLOv4 required approximately 32.0 seconds (average ~3.2 s/image). This direct comparison now clearly demonstrates YOLOv4’s slower inference in our setup, which contrasts with common expectations for a one-stage detector and underscores the importance of framework-specific implementation details.

In the Discussion section (“One-stage vs two-stage detection”), we expanded the discussion to highlight that while YOLOv4 is theoretically expected to be faster due to its one-stage architecture and optimized Darknet backbone, real-world performance can diverge depending on the software framework, hardware utilization, and post-processing strategies. In contrast, Faster R-CNN’s more stable and mature implementation in the PyTorch ecosystem may explain its relatively faster inference in our tests. These findings emphasize the necessity of conducting benchmarking under consistent, real-world conditions, as actual runtime performance may differ significantly from theoretical assumptions.

We also now cite two relevant studies [48, 49] that have reported similar results, where YOLOv4 (or relevant Darknet-53 extractor) demonstrated slower inference performance than Faster R-CNN (or relevant ResNet-50 extractor) in certain contexts, further supporting the observed discrepancy.

Editor Comment 6: Classification Metrics Section Too Lengthy:

“The section describing classification metrics is disproportionately long and remains overly didactic. While the inclusion of metric definitions is acceptable, the full mathematical formulas may be better suited to supplementary material, allowing the Methods section to remain focused and concise.”

Response:

We appreciate the editor’s suggestion and have revised the Classification Metrics subsection in the Methods (page 10-11, lines 197-204) to improve clarity and conciseness. While retaining essential definitions for precision, recall, F1-score, Intersection over Union (IoU), and mean Average Precision (mAP), we have relocated the full mathematical formulas and extended explanations to the Supplementary Materials (Table S4). This adjustment ensures that the main text remains focused and accessible, in line with journal expectations for readability and methodological clarity.

Editor Comment 7:

“Moreover, the process of image annotation and class distinction requires clarification. The ms must specify whether the distinction between visually similar classes (e.g. mud drapes vs bioturbated muddy media) was based on consistent quantitative thresholds, expert visual assessment, or a combination of both. The current description remains vague and limits reproducibility.”

Response to Editor:

We appreciate the editor's request for clarification. The manuscript has been revised to explicitly describe the annotation protocol used for visually similar classes. In Discussion section (“Challenges Related to Labeling Inconsistency”) (page 33, lines 635-642), we now state that particular attention was given to abundant and frequently misclassified classes, such as mud drapes and bioturbated media. These were consistently annotated using sedimentological criteria informed by expert visual assessment. Specifically, mud drapes were identified as fine-grained, laterally continuous to semi-continuous layers recurring in heterolithic facies, while bioturbated media were characterized by irregular disruption and burrow textures in muddy or sandy matrices. These distinctions were guided by domain expertise to ensure reproducibility and minimize ambiguity. Additionally, we refer to this approach in the “Sedimentary Structures” subsection, reinforcing the methodological rigor of the labeling process.

Editor Comment 8: Lack of formal comparison with prior work (e.g., Zhang 2021)

“Despite the claim that this is the first large-scale application of object detection to sedimentary structures, no formal comparison with previous studies is provided. The authors are required to include a summary table comparing their dataset, class range, and performance metrics with prior works, including Zhang 2021 and studies applying CNNs to lithofacies identification.”

Response to Editor:

We appreciate this valuable comment. To contextualize the contribution of our study within the existing literature, we have now included a direct comparison with Zhang et al. (2021), one of the only prior studies applying CNNs to sedimentary structures. The comparative summary (now provided in Table 6) outlines differences in dataset scale, class diversity, validation strategy, model type, and output granularity. We also integrated a concise synthesis of this comparison in the opening paragraph of the Discussion section (page 26, lines 490-499), where we highlight how our work extends prior efforts through the use of object detection, multi-class bounding box annotations, and structured split validation. This addition helps underscore the novelty and scale of our approach relative to earlier classification-based studies.

Editor Comment 9: Figure Captions:

“In addition, the figure captions should be carefully revised and substantially shortened. Several current captions (e.g. Figures 5 to 13) are excessively detailed and contain interpretative content or repetitions of information already presented in the main text. Figure legends should be limited to concise descriptions of what is shown...”

Response to Editor:

We thank the editor for this constructive feedback. In accordance with the journal’s guidelines, we have carefully revised the captions for Figures 5 to 13 to ensure brevity and clarity. All interpretive content, extended model comparisons, and data analysis previously embedded in figure legends have been relocated to the Results or Discussion sections as appropriate. The revised captions now succinctly describe each figure and its panels, maintaining only essential explanatory information (e.g., class names or key identifiers) necessary for visual interpretation. This streamlining ensures consistency with journal standards and enhances the manuscript's readability.

Editor Comment 10: Language Clarity and Redundancy:

“Finally, while the English language is generally clear, there are residual issues with fluency, repetition, and phrasing. Phrasal redundancy (e.g. ‘the model demonstrates ability to learn’ or ‘predictions show confusion between…’) appears frequently and should be eliminated. A thorough proofreading for academic tone and conciseness is required.”

Response to Editor:

We thank the editor for this helpful comment. In response, we have thoroughly revised the manuscript to eliminate phrasal redundancies, improve fluency, and ensure a concise and academically appropriate tone throughout. This has been applied in the clean copy not in the track-changes copy. Particular care was taken to revise expressions such as “the model demonstrates ability to learn” and “predictions show confusion between…” along with similar instances, aligning the text with the journal’s standards for clarity and style.

---

## [Editor Report · Decision Letter 2]

Automated identification of sedimentary structures in core images using object detection algorithms

PONE-D-25-08754R2

Dear Dr. Abdlmutalib,

We’re pleased to inform you that your manuscript has been judged scientifically suitable for publication and will be formally accepted for publication once it meets all outstanding technical requirements.

Kind regards,

Przemysław Mroczek, Dr. hab.

Academic Editor

PLOS ONE

Additional Editor Comments (optional):

Dear Authors,

Thank you for your thorough and constructive revision of the manuscript. I appreciate the substantial effort you invested in addressing all editorial and reviewer comments. The revised version demonstrates clear improvement in both scientific transparency and presentation quality.

You have successfully implemented all mandatory revisions. The manuscript now includes clear justification for expert validation, a well-documented approach to class imbalance, an improved discussion of misclassification errors, a detailed comparison with prior works, concise figure captions, and more focused classification metric descriptions. Supplementary tables and inference benchmarking have also been comprehensively provided.

I noted a few remaining minor language issues, such as occasional redundant phrases ("the model demonstrates ability to learn"), repeated sentence structures ("This study shows..."), or non-native phrasing (e.g., missing articles in some places, slight tense mismatches). These issues are minor and do not impact the scientific clarity of the manuscript; they will be polished during the journal’s professional copyediting stage.

Overall, your work meets PLOS ONE's publication criteria and is a valuable contribution to the field.

I am pleased to recommend acceptance of your manuscript for publication in PLOS ONE.

Best regards,

Przemysław Mroczek, Ph.D.

Academic Editor
---

## [Editor Report · Acceptance letter]

PONE-D-25-08754R2

PLOS ONE

Dear Dr. Abdlmutalib,

I'm pleased to inform you that your manuscript has been deemed suitable for publication in PLOS ONE. Congratulations! Your manuscript is now being handed over to our production team.

Kind regards,

on behalf of

Dr. hab. Przemysław Mroczek

Academic Editor

PLOS ONE